# The Coastline Evolution Model 2D (CEM2D) V1.1

Chloe Leach[1], Tom Coulthard[2], Andrew Barkwith[3], Daniel R. Parsons[2], Susan Manson[4]

[1]School of Geography, The University of Melbourne, Parkville, Melbourne, VIC 3010, Australia
[2]Department of Geography, Geology and Environmental Science, University of Hull, Hull, HU6 7RX, UK
[3]British Geological Survey, Environmental Science Centre, Keyworth, Nottingham, NG12 5GG, UK
[4]Environment Agency, Crosskill House, Mill Lane, Beverley, HU17 9JW, UK

*Correspondence to*: Chloe Leach (chloe.leach@unimelb.edu.au)

**Abstract.** Coasts are among the most intensely used environments on the planet, but they also present dynamic and unique
hazards including flooding and erosion. Sea level rise and changing wave climates will alter patterns of erosion and
deposition, but some existing coastline evolution models are unable to simulate these effects due to their one-dimensional
representation of the systems, or of sediment transport processes. In this paper, the development and application of the
Coastline Evolution Model 2D (CEM2D) are presented, that incorporates these influences. The model has been developed
from the established CEM model and is capable of simulating fundamental cause-effect relationships in coastal systems. The
two-dimensional storage and transport of sediment in CEM2D, which is only done in one-dimension in CEM, means it is
also capable of exploring the influence of a variable water level on sediment transport and the formation and evolution of
morphological features and landforms at the mesoscale. The model sits between one-dimensional and three-dimensional
models, with the advantage of increased complexity and detail in model outputs compared to the former, but with more
efficiency and less computational expense than the latter.

**1 Introduction**

Coastal systems are amongst the most dynamic environments on the planet, with their form and evolution being highly
sensitive to changes in environmental conditions, over a range of spatial and temporal scales (Wong et al., 2014). Under the
context of rising global sea levels and considering the social and economic importance of many coastal locations,
understanding the behaviour and potential future evolution of coastal environments is essential for the development of
suitable and sustainable management (Wong et al., 2014). Numerical models are increasingly being used for this purpose,
providing powerful tools that can give an insight into the complex morphodynamics and sensitivities of coastal systems (e.g.
Ashton, Murray and Arnault, 2001; Nam, Larson, Hanson and Hoan, 2009; Nicholls et al., 2012).

Simulating changes in coastal geomorphology up to millennial timescales and up to hundreds of kilometres, herein referred
to as the mesoscale, is highly relevant for coastal management and also fits with our historic frame of observation for model
validation and calibration (French et al., 2015; van Maanen et al., 2016). This scale sits between reduced complexity

reductionist studies and complex synthesist investigations, which have more traditionally been the focus of research into coastal behaviours (Fig. 1) (van Maanen et al., 2016).

Reductionist or 'bottom-up' models are designed to investigate small scale processes that act over relatively short timescales (Fig. 1) (van Maanen et al., 2016). They typically simulate complex behaviours by including a large range of processes that could influence the evolution of the system using more detailed calculations at higher resolutions (van Maanen et al., 2016). Using these types of models for mesoscale applications would be computationally expensive and inefficient, since there are a large number of processes that could be simulated over relatively long time scales (van Maanen et al., 2016). Decisions

would have to be made about which processes to include, since each process adds computational expense and additional uncertainty, which can propagate errors or inaccuracies over long simulated timescales (Hutton, 2012; Murray, 2007). Mesoscale models, like many types of model, should be parsimonious and include only fundamental processes that capture the main physical dynamics of a system, thus minimising model uncertainty (Wainwright and Mulligan, 2013).

Synthesist or 'top-down' models are designed to simulate large scale behaviours that act over longer time periods and often include only a few parameterised processes (Fig. 1) (Murray, 2007; van Maanen et al., 2016). They are intended to represent general behaviours and patterns in natural systems, rather than pertaining to spatially explicit research questions (Murray, 2007). As such, synthesist models are relatively limited in their ability to provide a level of understanding and prediction of coastal behaviours that is required for mesoscale research (Murray, 2007).


In the field of coastal modelling, there is a gap for a two-dimensional coastal model that can simulate features such as spits, bars and beach migration along with a dynamic nearshore bathymetry and a variable water level but is parsimonious enough to enable short run times allowing us to answer research questions about coastal evolution at meso-spatiotemporal scales. Existing models with such scope, such as CEM, COVE and GENESIS (Hanson and Kraus, 1989; Ashton et al., 2001; Hurst

et al., 2014), are limited to transporting sediment in one-dimension and represent the coastline simply as a line with little accommodation for the nearshore shape or bathymetry. This means the models are parsimonious and fast, but are limited in their application, for example, to investigate the effects of sea level rise on costal geomorphology. Hybrid shoreline change models such as COCOONED (Antolínez et al., 2019) and CoSMoS-COAST (Vitousek et al., 2017) calculate sediment transport in cross-shore and long-shore directions and can varying the water level in model but are transect based and do not

include a dynamically evolving bathymetry. The LX-Shore model (Robinet et al., 2018) is cellular-based with longshore and cross-shore sediment transport calculations but has an equilibrium beach profile as in models such as CEM and COVE. In contrast to these longer-term models, finer scale models such as Delft3D (Lesser et al., 2004) can simulate coastal hydrodynamics and sediment transport processes in two- or three-dimensions, but their complexity and long model run times means investigating sea level rise responses over meso-timescales is presently impracticable.


In this paper, the development and application of the Coastline Evolution Model 2D (CEM2D) is presented. This model is based on the underlying assumptions of the CEM, but with sediment transport processes that are applied over the two-dimensional grid, which allows us to represent the morphology of coastlines in more detail and incorporate sea level rise. A key aim of the model development is to create a tool to improve our understanding of the mesoscale morphodynamic behaviour of coastal systems, their sensitivities and the influence that sea level rise may have on their evolution over centennial to millennial timescales. We describe in full the model's operation and parameterisation, and compare the model outputs to the original CEM, illustrating some similarities in model outputs but also key differences that are due to the improved two-dimensional representation of the coastline and sediment transport processes. Validation of exploratory models like CEM2D is limited and particularly in this case, where there is a lack of data showing the evolution of coastal systems under changing wave patterns and water levels over such long time periods. CEM2D's performance is therefore here evaluated against 'standard' CEM simulation results from varying wave climates and directions.

## 2 The Coastline Evolution Model (CEM)

As CEM2D builds on many concepts developed in the original CEM, it is important to first understand how CEM operates. CEM is grid based, dividing a plan-view coastline into a grid of regular square cells, of a user-defined size (m). Each of these cells contains a fractional proportion of sediment (Fi) that represent its horizontal fill across the domain. The Fi values are updated according to the longshore transport of sediment and the landward or seaward migration of the shore (Ashton et al., 2001). Cells can be defined as fast or slow eroding, to represent basic lithological characteristics of a coastline.

The one-line coastline can be drawn along shoreline cells, at the interface between land and sea cells. A shoreline search technique is used to locate these shoreline cells. The initial shoreline cell on the left side of the domain is located by iterating through the first column of cells from the top down, until a land cell is found. A clockwise search is then used around the first shoreline cell to locate the next cell. This is then repeated until all shoreline cells are found. The angle of the deep-water wave crest and local shoreline orientation determines the direction of sediment transport between cells. If the local relative wave angle is less (greater) than the angle which maximises sediment transport, sediment flux is calculated using a central (upwind) finite-difference technique (Ashton et al., 2001; Ashton and Murray, 2006a).

The sediment flux and net erosion or accretion of material in each cell determines the cross-shore movement of the shoreline and is controlled by wave-induced sediment transport calculated using the CERC formula in terms of breaking wave quantities following Eq. (1):

$$Q_s = K H_b^{\frac{5}{2}} sin(\phi_b - \theta) cos(\phi_b - \theta) \tag{1}$$

here, Qs is the sediment flux (m³/day), K is a calibration coefficient, Hb is the breaking wave height (m), $\phi_b$ is the breaking wave angle (°) and θ is the local shoreline orientation (°). Breaking wave characteristics are calculated from an offshore wave climate that is transformed over assumed shore-parallel contours, using Linear Wave Theory (Ashton et al., 2001). An arbitrary offshore water depth is iteratively reduced and the offshore wave angle and height recalculated until the waves break. The wave climate characteristics at the point of breaking are then used to compute the sediment flux between each cell and the net erosion or deposition of sediment using Eq. (2) (Ashton et al., 2001):

$$\Delta\text{Fi} = Q_{s,net}\Delta\text{t}/(W^2 D_i) \tag{2}$$

where W is the cell width and Di the depth to which significant sediment transport occurs, known as the Depth of Closure (DoC). The DoC is defined as the location from the shore where the depth of water is greater than the depth of wave influence and therefore, the flow has a negligible impact on cross-shore sediment transport; this depth is often approximated as half the average wavelength (Hallermeier, 1978; Nicholls et al., 1997; Pinet, 2011). The assumed location of the DoC in CEM is the point where the continental shelf and the linear shoreface slope intersect (Fig. 2) (Ashton et al., 2001). The slope of the shoreface is assumed constant and does not evolve morphologically throughout simulations or vary the beach profile. Sediment is not transported out of cells that are shadowed by protruding sections of coastline, since they are protected from incoming waves (Fig. 3).

Where a shoreline cell overfills with sediment (Fi > 1), the excess material is deposited in the surrounding empty cells. As new cells become active land cells, the shoreline advances. This redistribution of material has no effect on the topographic profile of the coastline, but simply shifts the location of the shoreline to where cells have filled with sediment. If a greater volume of sediment is removed from a cell than it contains, the shoreline retreats. With this one-line approach, effectively the water level in the model is held constant and cannot be varied, which limits its application to studies interested in the influence of sea level change on coastal evolution.

**3 The Coastline Evolution Model 2D (CEM2D)**

CEM2D contains a significant number of modifications to enable it to model the evolution of coastal features including their topographic profiles and to study the influence of a variable water level. The model domain is divided into regular square cells of a user-defined size (m), as per CEM (Fig. 4a). The variable Fi is not used in CEM2D to represent the partial horizontal fill of sediment, rather, each cell contains values for depth of sediment to the continental shelf, elevation of sediment above the water level or depth of water (Fig. 4b). Having these additional values of sediment fill in the vertical enables CEM2D to represent two-dimensional coastlines with greater topographic detail compared to the original CEM, as illustrated in Fig. 4. Importantly, the two-dimensional profile allows the morphology of the beach and shoreface to evolve

according to the transport of sediment, across the entire model domain. It explicitly models the slope of the continental shelf and shoreface and the morphological profile of the beach and sea floor.

In CEM2D the elevation of each cell relative to the water level is used to classify cells as either wet or dry on each model iteration. The boundary between wet and dry cells is used to locate the shoreline (Fig. 5) using the same shoreline search technique as CEM. The local shoreline orientation is identified by computing the angle between a shoreline cell and two neighbouring shoreline cells. This forces the shoreline angle to be either 0, 22.5, 45, 67.5 or 90 degrees. Once the shoreline is located and the local shoreline angle computed, as per CEM, Linear Wave Theory is used to transform the offshore wave climate and the CERC formula to calculate sediment flux between the one-line shoreline cells (Equation 1). The limitations of not calculating the horizontal sediment fill of each cell (Fi) influences the sediment transport equations by reducing the angular resolution of the local shoreline and it can also lead to a more irregular representation of the shoreline. However, as shown in the results, the model remains capable of simulating fundamental shoreline shapes.

Sediment flux is calculated using the same equations as CEM (Eq. 1), employing threshold-determined upwind or central finite-difference techniques (Ashton et al., 2001; Ashton and Murray, 2006a). However, since CEM2D represents sediment transport in two-dimensions, an alternate method for distributing sediment across the surf zone is used. Rather than assuming shore-parallel contours, material is dispersed across the surf zone based on an avalanching scheme that is somewhat similar to that used in other coastal evolution models (e.g. XBeach (Roelvink *et al.*, 2009))(Fig. 6). The method ensures that sediment is distributed across the active profile and remains consistent with transport calculations using the integrated CERC formula, but that there is dynamism in this process that takes into account the elevation of a cell and its neighbours that is consistent with the 2D representation of the domain.

The sediment distribution method is based on the relationship between the properties of coastal material (e.g. sand, gravel) and slope angle, as shown by McLean and Kirk (1969). We can assume that in general, coastal profiles will maintain an average slope angle consistent with the grain size of beach material, although there are a range of factors that can cause steepening or shallowing (McLean and Kirk, 1969). To carry out this redistribution procedure, an algorithm sweeps the entire model domain and identifies where a critical angle has been exceeded between a cell and its neighbour (Eq. 3).

$$\frac{\Delta_z}{\Delta_w} > m_{cr} \tag{3}$$

where z is depth, w is cell width and $m_{cr}$ is the critical slope. The material is then redistributed amongst the orthogonal surrounding cells until the critical slope angle is no longer exceeded (Fig. 6).

The sediment metrics are then updated accordingly, including the total volume of material and the cell's elevation above a reference point. The rules defining the sediment redistribution are important parameters that can significantly alter the model outcomes and have therefore been thoroughly tested. The two most critical components are (1) the threshold angle between

cells that instigates transport and (2) the frequency that the domain is analysed for these thresholds. These values should be calibrated to allow sediment to be distributed without inducing sediment pilling or deep depressions forming in the domain. Similar techniques are widely implemented in landscape evolution models, such as SIBERIA (Willgoose et al., 1991) and GOLEM (Tucker and Slingerland, 1994) (Coulthard, 2001). The implementation of this method in CEM2D allows the nearshore profile to evolve dynamically, rather than assuming an even distribution across the nearshore profile and forming shore-parallel contours, as is the case in CEM and other one-line models. In CEM2D, the ability of the simulated coast to evolve dynamically in this way provides a more realistic representation of the morphodynamic behaviour of these systems. How sediment is distributed can affect the longer-term evolution of the system and record a morphological memory of landforms which can interact with other features as they form and mature (Thomas et al., 2016).

CEM2D's two-dimensional structure allows the water level to be varied, but by default the water level is at 0 m elevation. There are two dynamic water level modes within the model which can be run independently or in combination that can be used to represent tidal fluctuations and long-term sea level change. The increased complexity of the model domain and of sediment transport processes in CEM2D enable it to model complex two-dimensional coastal profiles and evolve their morphology. The features allow more complex morphodynamic processes to be explored and to investigate not only the evolution of the one-line shore, but the surrounding beach and shoreface. The sediment storage and handling technique allow complex landforms and features to develop and leave a morphological memory in the bathymetry as they evolve. Sea level change is an important addition to this model that could be used to explore the response of coastal systems to fluctuating water levels and the influence of fundamental climate change effects such as sea level rise.

## 4 Methodology: Sensitivity Analysis and Model Evaluation

To evaluate how CEM2D simulates coastal change, CEM2D was compared to CEM model outputs as well as to the behaviour and morphology of natural coastal environments. This provides both a check that the new model is able to represent natural systems as the original, but also to indicate where the added features (namely 2D operation) might change the model outputs. As the aim of this paper is to describe and highlight the technical developments of CEM2D we evaluate our simulation results against the original CEM outputs (as described subsequently). Full validation would require time series of bathymetric field data for the duration and range of wave climates and wave directions simulated, and this is not presently available. However, similar to how (Ashton and Murray, 2006) visually compare their simulation findings to coastal features including the Carolina Capes, we too compare our outputs to a series of examples.

### 4.1 Initial Conditions

CEM and CEM2D were initially set up with a uniform gridded domain measuring 200 (cross-shore) by 600 (longshore) cells, with a cell size of 100 m by 100 m (Fig. 7). A straight planform coastline was used, with uniform undulations along its

length. The coastal profile is characterised by a fixed continental shelf slope of 0.1 with a minimum imposed depth of 10 m and an average shoreface slope of 0.01. Within CEM2D, these average slopes are imposed across the two-dimensional domain including the beach and bathymetric profiles which are built to replicate an average coastal profile slope of 0.01. The left and right boundaries of both model domains are governed by periodic boundary conditions, to allow a constant flux of sediment from one end to the other and conserve the volume of material in the system. No-flow conditions were set at the seaward end of the domain to again, conserve sediment and prevent any gain or loss of material. A daily model time step is used for all simulations. The models were run over a simulated period of 3,000 years, to allow time for the model to spin-up (~10 years), to reduce the potential influence of initial conditions and to allow sufficient time for the coastal systems to evolve.

### 4.2 Wave Climate Conditions

An ensemble of wave climates was used to drive the model in order to explore the influence of wave conditions on the morphology and evolution of coastal systems. We use the four binned Probability Density Function (PDF) approach of Ashton and Murray (2006a) to define the proportional asymmetry (A) of waves and the proportion of high angle waves (U) approaching the coastline, according to the wave crest relative to the average shoreline orientation (Fig. 8). Twenty-five simulations were completed, with A values varying between 0.5 and 0.9 in increments of 0.1 and U values that varied from 0.55 to 0.75 at 0.05 increments. The pseudo-random wave angle was generated on each iteration, according to these proportional values. The wave height and period are held constant, at 1.7 m and 8 s respectively.

### 4.3 Water Level

The primary purpose of this paper is to highlight the technical development of CEM2D and demonstrate its additional functionalities. The simulations shown focus on how the coastal systems evolve with an unchanging water level at 0 m elevation, but results are also given for how an increasing water level at a rate of 2 m / 100 years influences the evolution of four shoreline types: cuspate, sand wave, reconnecting spit and flying spit. This rate of rise is in line with the UK Climate Projections 2009 (UKCP09) (Jenkins et al., 2009) H++ scenario of 0.93-1.9 m sea level rise by 2100 (Jenkins et al., 2009; Lowe et al., 2009).

### 4.4 CEM2D Sensitivity Analysis

A sensitivity analysis (SA) technique designed by Morris (1991), and subsequently adapted by Campolongo et al., (2007), was used to identify the relationship between model inputs and outputs by performing multiple local SAs to approximate model sensitivity across a global parameter space. The Morris Method's design of experiment uses a defined set of values for each input factor, which are discretised into equal intervals and constrained by upper and lower boundaries (Morris, 1991; Ziliani et al., 2013). Each value is altered incrementally per model sensitivity simulation and the elementary effect of each factor on model outputs is calculated according to the variance of performance indices, by Eq. (4):

$$d_{ij} = \left( \frac{y(x_1 x_2 \ldots x_{i-1}, \ x_i + \Delta_i, \ x_{i+1}, \ldots, x_k) - y(x_1 x_2 \ldots x_{i-1}, \ x_i, \ x_{i+1}, \ldots, x_k)}{\Delta_i} \right) \hspace{2cm} (4)$$

where $d_{ij}$ denotes the value of the *j*-th elementary effect ($j = 1,\ldots,r$) of the *i*-th input factor (and where r is the number of repetitions), $y(x_1 x_2,\ldots,x_k)$ is the value of the performance measure, k is the number of factors investigated and $\Delta$ is the incremental step value. The main effect is then calculated according to the mean (μ) of multiple elementary effects computed randomly from the parameter space, which indicates the relative influence of each input factor on model outputs (Ziliani et al., 2013). The standard deviation (б) is also used to determine which, if any, input factors have nonlinear effects or which have an influence on model output but in combination with other unspecified inputs (Ziliani et al., 2013).

The number of input factors tests and the number of repeats using the Morris Method was constrained by resource availability and computational expense. Further, as demonstrated by Skinner et al., (2018) behavioural indices can be used in the place of performance indices where there is a lack of data to populate the performance indices to drive a more qualitative assessment of model sensitivity. A total of eight key input factors were tested against four behavioural indices that represented fundamental processes in the model. The input factors were each ranked according to their relative influence on model outputs and to determine which, if any, input factors have nonlinear effects or which have an influence on model output but in combination with other unspecified inputs (Ziliani et al., 2013). The factors tested are given in Table 1 and the behavioural indices in Table 2.

## 5 Results

### 5.1 CEM2D Sensitivity Analysis

The mean and standard deviation of each input factor on each behavioural index is given in Fig. 9. The higher the mean, the greater the influence of that factor on model outputs and the higher the standard deviation, the greater the nonlinearity; nonlinearity refers to the nonsequential effects of the given factor on model sensitivity or that it influences model behaviour through complex input-input interactions (Ziliani et al., 2013; Skinner et al., 2018). The results show the principle input factors which (1) have the greatest influence on model sensitivity (e.g. wave angle, wave height, sediment distribution factors), (2) those which have a negligible influence (e.g. wave period and domain characteristics) and (3) those which show nonlinear behaviours or interactions which can amplify variance in model outputs (those which also have the greatest influence on model behaviour, e.g. the wave angle). The results further highlight input factors that can have an influence on model outputs, but only according to specific behavioural indices (e.g. water level and domain characteristics). It is important to note that the results of the SA can be influenced by the input factors used, the range of values and the behavioural indices that are chosen to assess sensitivity.

Aggregating the results from the four behavioural indices shows that the wave angle and height have the highest-ranking influence on model behaviours, followed by sediment distribution factors and the domain set-up is considered the least influential (Fig. 9). Factors which rank highly based on the mean, also tend to show greater nonlinearity and have complex interactions with other inputs. It is also found, however, that the rankings of the various input factors differ according to the behavioural indices used to assess model sensitivity, each of which describes a different behaviour in the model. For instance, the water level shows a high influence on model behaviour when assessed against the ratio of wet to dry cells but according to the sinuosity of the shoreline, is ranked just below average. The selection of model parameters, described in methods, was driven by the results of the SA and particular attention was given to constraining optimum wave climate conditions and sediment distribution parameters through a series of further behavioural sensitivity testing.

## 5.2 Fundamental Shoreline Features

The ensemble plots in Fig. 10 and Fig. 11 show final coastal morphologies produced from CEM and CEM2D respectively according to the twenty-five wave climate conditions. Both models demonstrate how different planform shoreline shapes evolve according to the wave climate scenarios, as previously demonstrated by Ashton and Murray (2006a). The proportion of high angle waves influences cross-shore sediment transport and the extent to which landforms accrete seaward, whilst the wave asymmetry determines the balance of cross- to longshore transport and the planform skew of features. It is found that there is some directional bias in the source code that drives a longshore current independent of the wave climate conditions. This directional bias is more apparent in CEM2D and particularly where the wave climate is symmetrical (A = 0.5). It also drives some migration of the cuspate landforms downdrift, but a similar rate of movement is recorded in both CEM and CEM2D at 1.6 m and 1.7 m per year respectively. The directional bias is induced by calculations in the model that process from the left to the right of the domain. In future model versions, the routines will require updating which would also necessitate that sediment transport methods be altered accordingly.

Four principle shoreline shapes evolve under the driving wave conditions including cuspate forelands, alongshore sand waves, reconnecting spits and flying spits. CEM2D shows a greater sensitivity to inputs variables compared to the CEM, apparent in the development of these four feature types. In CEM2D a greater distinction is made between reconnecting and flying spits due to the increased complexity of CEM2D's sediment handling and distribution methods. The distribution method allows sediment accumulations to be detached from the continuous shoreline without becoming static and so transport across the entire domain, including on the lee side of a spit, is less limited. Each of these four features types are compared to natural systems subsequently that are subject to comparable wave climate conditions.

### 5.2.1 Cuspate Forelands

Symmetrical wave climate conditions (A=0.5) are shown to form cuspate forelands in CEM and CEM2D, which compare to those found along many shorelines globally. The Carolina Capes span parts of North and South Carolina's coast in the USA and are used as a case site by Ashton and Murray (2006b) against results generated by CEM. The wave climate along this stretch of coastline is characterised by high angle waves of relative symmetry, which broadly equate to PDF values of A = 0.55 and U = 0.6 (Ashton and Murray, 2006b). Placing the Carolina Capes into the context of the results shown in Fig. 11, CEM2D would model a cuspate coastline which is slightly skewed due to the 5% dominance of left-approaching waves. The wave direction plays a significant role in the formation of the features, with the slightly stronger southerly current skewing the tips of the landforms (Park and Wells, 2005). Considering that all site-specific conditions controlling the evolution of capes are not represented in CEM2D or CEM, the models are able to predict a comparable shoreline type to that observed in this natural system. However, CEM2D overpredicts the directional skew and so CEM may be the preferred option in this instance.

### 5.2.2 Alongshore Sand Waves

A slight asymmetry in the wave climate (where A = 0.6) generates alongshore sand waves in both CEM (Fig. 10) and CEM2D (Fig. 11). However, CEM2D has a greater sensitivity to this parameter and the features show a greater skew downdrift. For instance, under A=0.6 and U=0.75 cuspate sand waves form along the shoreline in CEM, but in CEM2D the features skew and hooks form at the distal points. Comparing these results to the planform morphology of sand waves found in natural systems, such as Benacre Ness in the UK which has PDF values of A=0.6 and U=0.8, demonstrates the ability of CEM2D to reflect the asymmetry of landforms formed under asymmetric wave climate conditions compared to CEM; CEM2D may, therefore, be the preferred model in this instance. However, it is noted that site-specific environmental and boundary conditions play a role in the formation and evolution of Benacre Ness which are not modelled by either software and that the wave transformation equations used may not be wholly suited to this site.

### 5.2.3 Reconnecting and Flying Spits

Under high asymmetric wave climate conditions, dominated by high angle waves, spits forms along the shoreline in CEM (Ashton and Murray, 2001; Ashton et al., 2006b) and CEM2D. However, CEM2D again shows a greater sensitivity to the wave climate conditions, with more distinction made between reconnecting and flying spits due to the refinement of sediment handling techniques in the model.

Ashton and Murray (2006b) compare results from CEM to the behaviour and development of the reconnecting Long Point Spit in Lake Erie, Canada, where the wave climate is characterised by high asymmetry (A = 0.8-0.9) and high angle wave dominance (U = 0.6-0.7) (Ashton and Murray, 2006b). Under all four potential wave climate conditions, reconnecting spit

features form in CEM (Fig. 10), whereas in CEM2D (Fig. 11) either sand waves or reconnecting spits form depending on the combination of A and U values within the given ranges. Ashton and Murray (2007) suggest that the wave climate is favoured towards an asymmetry (A) of 0.8 along the entire spit and under these conditions, reconnecting spits form in CEM2D (Fig. 11), suggesting that CEM2D may be there preferred tool to use in this conditions. The presentation of both sand waves and reconnecting spits in CEM2D would suggest that this model may be able to better represent the conditions found at Long Point Spit.

Comparing model results to flying spits, Spurn Point in the UK extends off the southern end of the Holderness Coast and has a PDF wave climate of A = 0.75, U = 0.35. Following the pattern of results from CEM (Fig. 10) and CEM2D (Fig. 11), where there is proportional asymmetry (A) of between 0.7 and 0.8, net longshore sediment transport forms these types of landforms. However, in CEM2D these features fluctuate between spits and sand waves owing to the strong longshore current generated by the low angle waves and high asymmetry. Whilst CEM2D better represents the influence of low angle waves on coastal evolution at Spurn Point, it is of note that this is a complex feature which is influenced by conditions that could be having a greater impact on coastal evolution, including estuarine processes and dredging activities, that are not included in either CEM or CEM2D.

### 5.3 Spatial Scale of Shoreline Features

The spatial scale of shoreline features differs between results from CEM and CEM2D. Metrics from the end of each run, as shown in Fig. 10 and Fig. 11, show larger features evolve in CEM2D in six of the simulations. The larger features evolve under wave climate conditions where A = 0.6, U = 0.55-0.65 (sand waves), where A = 0.7-0.8, U = 0.7 (flying spit) and where A = 0.9, U = 0.75 (flying spit) and smaller features evolve in the remaining nineteen simulations. However, each run terminates at a different timestep and a comparison of results at the earliest termination for each pair of simulations shows that in all but one of the runs (A = 0.6, U = 0.55), the features are smaller and less developed in CEM2D than the CEM (Fig. 12).

Whilst CEM and CEM2D are not designed to represent the temporal evolution of specific coastal environments and this metric should not therefore be compared between the models (Ashton and Murray, 2006a), we note that the evolution of landforms is more gradual in CEM2D. This is likely as a result of differences in the representation of the domain and in the distribution of sediment. Rather than sediment being distributed evenly across the nearshore to the depth of closure, as in CEM, CEM2D uses the sediment distribution method to route sediment along lines of steepest descent and spreads available material across the nearshore profile. This leads to both the formation of shoreline features but also to the formation of a shallow nearshore shelf (see Section 5.4).

Highlighted above are differences in results between CEM and CEM2D and in particular, the complexity of results generated

in CEM2D due to the addition of a dynamically evolving profile. In nature, the features discussed evolve at different rates and to different spatial scales depending and in order to use CEM2D to investigate such systems, parameters in the model including the threshold and frequency of sediment distribution should be adjusted to suit the specific environment studied and the rates at which these features form.

### 5.4 Dynamic Coastal Profile

The novel development of CEM2D is to simulate variations in the nearshore topography. Of particular interest are the dynamics of the upper nearshore which evolves under the influence of sediment exchange with the shoreline (Fig. 13b). The lower nearshore profile tends to be influenced to a lesser degree (Fig. 13c) and consequently, is able to store remnants of morphological features as they evolve.

One-line models tend to assume that contours lie parallel to the shoreline, but the results in this study demonstrate that the bathymetric profile in particular is highly dynamic (Fig. 13). Whilst some of the results of CEM2D show a profile with shore-parallel contours, the majority do not exhibit this behaviour, particularly where there is a strong asymmetry in the wave climate (Fig. 13). The shoreline and bathymetry is not solely influenced by current environmental conditions but previous states and morphological residuals. Omitting or smoothing the bathymetry in the representation of coastal systems

could have implications for their long-term evolution. The effect of morphological inheritances have been previously suggested by authors including Wright and Short (1984), French et al., (2015) and Thomas et al., (2016). Many of the results from CEM2D have noted the presence of remnant features or states in the coastal profile, particularly in the nearshore zone. The presence of these features is strongly attributed to the balance of cross- and long-shore sediment transport, and the rate of change. For instance, where sand waves form the rate of change is such that the longshore movement of landforms makes

an impression in the profile that is significant enough to be sustained in the bathymetry as the features migrate (Fig. 14). However, where reconnecting spits form along the shoreline, the rapid rate of longshore and cross-shore sediment transport act to smooth the profile and remove evidence of predeceasing morphologies (Fig. 14). These processes could prove important for understanding the nearshore dynamics of natural coastal environments, particularly under changing environmental conditions.

Relative rates of morphological change and coastal dynamics differs according to the driving wave conditions (Fig. 15). This is illustrated in the volume stacks in Fig. 15 which present the change in volume of sediment across a transect (x = 30 km) every 30 simulated years, for four wave climate scenarios where (a) A = 0.5, U = 0.55, (b) A = 0.6, U = 0.6, (c) A = 0.7, U = 0.65 and (d) A = 0.8, U = 0.7. With increasing wave asymmetry and proportions of high angle waves, the active cross-shore

zone exhibits greater dynamism and greater volumes of net longshore transport. However, the results also show that these systems have complex non-linear behaviours that emerge from the balance of longshore and cross-shore sediment transport.

## 5.5 Variable Water Level

Changing the water level against the dynamic topography allows CEM2D to explore how a rising water level might affect how coastal systems behave. The results demonstrate that a rising sea level causes landward recession of the shoreline and uplift of the profile (Fig. 16), as is commonly held (Dickson et al., 2007; Bird, 2011). The rate of recession is broadly within two orders of magnitude the rate of sea level rise, prescribed at 2 m / 100 years in the simulations, which is in agreement with Bruun Rule estimations (Bruun, 1962). Variations in the rate of recession and morphology of the cross-shore profile are, however, observed with different wave climate conditions that differ in the balance of cross-shore and longshore flows (Fig. 16).

As in Fig. 15, Fig. 17 shows the change in volume of sediment across a transect (x = 30 km) every 30 simulated years, for four wave climate scenarios where (a-b) A = 0.5, U = 0.55, (c-d) A = 0.6, U = 0.6, (e-f) A = 0.7, U = 0.65 and (g-h) A = 0.8, U = 0.7. The figure shows a comparison of the results with a static water level (top) and with a rate of sea level rise of 2 m / 100 years (bottom). The spatial extend of morphological change is more diverse and widespread when the systems are subject to sea level rise. The principal active zone also tracks backwards as the water level rises and the shoreline recedes.

A rising sea level influences the evolution of shoreline features that evolve in the model, including cusps (a), sand waves (b), reconnecting spits (c) and flying spits (d) (Fig. 18). As also shown in Fig. 18, recession of the shoreline is observed in all four coastal systems as the water level rises regardless of the wave climate conditions (also shown in Fig. 16). Where the wave climate is symmetrical, cuspate features form under a static water level but have a slight asymmetry under sea level rise conditions, where the direction bias in the model is exaggerated. The cusps extend further offshore where the bays between the headlands are eroded, increasing wave shadowing and hence, exaggerating the effects of the directional bias. A slight asymmetry in the wave climate forms sand waves along the shoreline (Fig. 18b), but submergence of these features under a rising sea level leads to the formation of a waterbody in the low-lying interior of the landform. Where the wave climate is defined by A = 0.7, U =0.65 (Fig. 18c), reconnecting spits form when the water level is static, but as the water level rises the pathways that reconnect the spit to the mainland are submerged. In Fig. 18d, flying spits are shown to evolve with and without sea level rise, where the wave climate is highly asymmetric. The difference between these two simulations is that under a rising water level, the flying spits keep pace with the migrating shoreline and also cycle through submergence and reformation, as they are drowned by the rising water, but new features are able to form due to the high rate of longshore sediment transport. Remnants of the submerged spits remain in the nearshore and promote the development of spits in these areas due to the shallower water and also influence the unique plan-form morphology of the features.

Observing the effects of sea level rise on coastal features, including their ability to migrate with the shoreline or how their morphology changes at this temporal scale is challenging. Evidence of submerged shorelines and landforms that formed

during transgressive periods can be removed by high energy waves and storm events, rapid migration of systems and by sediment transport that consumes or removes remnant features (Shaw et al., 2009). Notable submerged shorelines are found in the Bras d'Or Lakes, Nova Scotia, and are suggested to have been well-preserved by the rapid onset of sea level rise (Shaw et al., 2009). Tombolos, spits, cuspate forelands and barrier beaches are identifiable on multibeam sonar imagery in the Lakes down to -24 m, above the early Holocene water level at -25 m (Shaw, 2006). Evidence of enclosed bodies of water within cuspate forelands and the stranding of landforms at this lower sea level demonstrates in situ drowning and the preservation of landforms between -7 to -24 m evidences the ability of some landforms to migrate (Shaw, 2006; Shaw *et al.*, 2009). Barrier islands and spits in the Bras d'Or Lakes are also found to rebuild at the proximal end of previously submerged landforms (e.g. Dhu Point and West Settlement) or migrate landward to form cuspate barriers in response to rising water levels (e.g. Goose Pond) (Taylor and Shaw, 2002).

## 6 Discussion

The purpose of this study was to provide an overview of the development and application of CEM2D and its ability to represent coastal systems compared to other existing coastal evolution models of its kind. The behaviour of the model has been evaluated against results from the existing Coastline Evolution Model (CEM), upon which CEM2D has been built. Results have also been compared to accepted theories of coastal morphodynamics and to the behaviour of a number of natural coastal environments. These evaluation techniques have demonstrated that CEM2D is able to simulate shoreline instabilities in accordance with theories of high-angle wave instability, to mimic the behaviour of natural environments under given wave climate conditions and to generally reproduce the results of the original one-line CEM, although some differences are observed as discussed throughout (Ashton and Murray, 2006a). In particular, the results show that CEM2D shows increasing model performance with increasing wave asymmetry compared to CEM. This is likely due to its ability to handle detached sediment accumulations that form during the evolution of reconnecting and flying spits, under these wave conditions. It may, therefore, be more appropriate to use CEM2D over CEM when modelling environments with asymmetric wave climates, but CEM where wave approach is highly symmetrical. Overall, our results show the sensitivity of coastal systems to driving environmental conditions and in particular their response to changing wave climates which supports theories of high angle wave instability.

Importantly, restructuring and increasing the dimensionality of sediment transport in the model allows us to explore how the profile of the coastal system changes with the shape of the shoreline, as well as concepts such as morphological inheritance. Where this is considered particularly important or of interest, CEM2D would be the preferred model to use over CEM. In many one-line models, the cross-shore profile of the coastline is kept constant and it is assumed that its core geometric properties are retained over meso-spatiotemporal scales. Whilst this is a well-used concept, there are advantages to modelling the topography and bathymetry of the coastline and it is necessary if we are to model the effect of a variable water

level. For example, we can see that the nearshore evolves at a greater rate compared to the lower shoreface profile, supporting the theories of Stive and de Vriend (1995). The distribution of sediment across the profile is more transient towards the shore where the greatest volume of transport occurs. However, the geometry of the entire shoreface and the

geometric demand for sediment distribution means that material is moved to the lower shoreface over time, but at a relatively slower rate (Stive and de Vriend, 1995). Further, the topographic profile of coastal landforms is indicative of their formation and evolution, highlighting patterns in sedimentation and drift processes. Using CEM2D to model how this profile changes over time can inform the stability and future behaviour of features.

The longshore sediment transport equations in CEM2D are inherited from the CEM and currently do not take into consideration the water depth, or how far from the shore the waves break. Since the water depth can now be calculated within this 2D model, future developments of CEM2D will focus on a revision of the sediment transport equation and include a more suitable calculation that can take advantage of the increased complexity and added functionalities in CEM2D.

A key component of CEM2D is its variable water level, which offers an added advantage over the use of CEM particularly when considering the impacts of sea level rise over the timescales these models are intended for. If we are to explore coastal evolution over the mesoscale, being able to model the effect of rising sea levels is essential. Whilst we have not exhausted the uses of this function here, we have demonstrated its development and how it is facilitated in the model. The power of this tool is vast and will be particularly useful for coastal managers who must plan for the dynamic evolution of these system

over time periods that will be highly influenced by the effects of climate change.

## 7 Conclusion

Here we have presented the development of CEM2D from its one-line origins. We have described the structure of the model, outlined the governing mathematical equations, presented outputs from the sensitivity testing and evaluated CEM2D's ability to simulate the behaviour and evolution of coastal systems, by comparing against other model results, theories of

coastal evolution and natural systems. The results demonstrate the validity of the model by its ability to simulate fundamental coastal shapes as per CEM and in comparison to natural coastal systems. Using the added functionalities, we have also shown how CEM2D can be used to explore the two-dimensional behaviour and morphodynamic evolution of coastlines and depositional features, over meso-spatiotemporal scales. From the results shown here, it is apparent that the model will enable us to conduct interesting and insightful investigations to answer research questions including how coastal

systems behave under changing environmental conditions and how sea level change might influence their morphodynamic behaviour.

## 8 Code Availability

The current version of the Coastline Evolution Model 2D (CEM2D) is available from the project website: https://sourceforge.net/projects/coastline-evolution-model-2d/ and on Zenodo (DOI: 10.5281/zenodo.3341888) distributed
under the terms of the GNU General Public License.

## 9 Author Contributions

All Authors contributed to writing and editing this manuscript. The research and software development was led by CL, with extensive support from TC and AB.

## 10 Competing Interests

The authors declare that they have no conflict of interest.

## 11 Acknowledgements

This research was funded by the UK Environment Agency, the NERC funded British Geological Survey and by the Energy and Environment Institute at the University of Hull.

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

**Table 1: A table listing the eight input factors from CEM2D used in the sensitivity analysis of Morris Method.**

| Code | Factor | Intervals | Minimum | Maximum | Justification |
|---|---|---|---|---|---|
| 1 | Wave Angle (º) | 5 | 1 | 5 | The wave climate is fundamental to driving sediment transport processes in CEM2D. |
| 2 | Wave Height (m) | 5 | 1 | 6 | |
| 3 | Wave Period (s) | 5 | 1 | 14 | |
| 4 | Sediment Redistribution Frequency (iterations) | 5 | 10 | 50 | The sediment redistribution method is a new scheme in CEM2D, governed principally by factors which defined the frequency and threshold for sediment redistribution. |
| 5 | Sediment Redistribution Threshold (%) | 5 | 1 (%) | 100 (%) | |
| 6 | Water Level Change (m) | 5 | 0 | 2 | The ability to induce sea level rise in the model is a new scheme that requires testing for its influence on model outputs. |
| 7 | Initial Shoreline Shape | 3 | 1 | 3 | The original CEM claimed to be relatively insensitive to these initial conditions. Increasing the dimensionality and complexity of sediment transport in the model warrants that their influence on CEM2D outputs be evaluated. |
| 8 | Domain Width (km) | 3 | 1 | 3 | |

**Table 2: A table showing the 4 behavioural indices used in the Morris Method and the frequency that data is recorded in each simulation.**

| Number | Behavioural Index | Recording Frequency |
|---|---|---|
| 1 | Longshore sediment transport rate ($m^3$ / 10 years) | 3650 model iterations (10 simulated years) |
| 2 | Coastal sinuosity | 3650 model iterations (10 simulated years) |
| 3 | The ratio of wet-dry areas | 300 model iterations (300 simulated days, to align with each diffusion frequency tested) |
| 4 | Run duration (simulated years) | 1095000 model iterations (3,000 simulated years) |


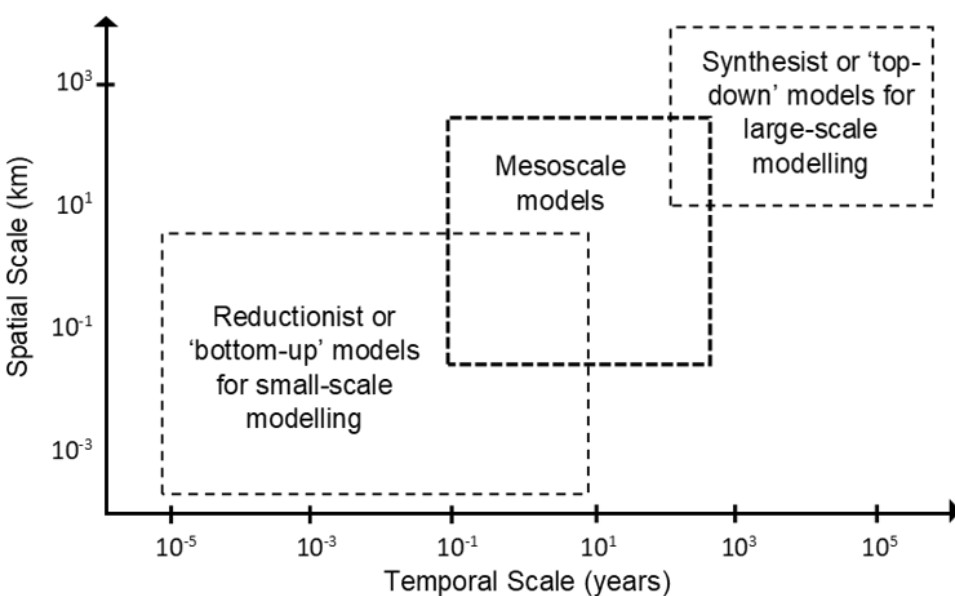

**Figure 1: Spatial and temporal ranges for traditionally reductionist and synthesist models, with mesoscale models highlighted in grey within the scale appropriate for coastal management (adapted from Gelfenbaum and Kaminsky, 2010; van Maanen et al., 2016).**

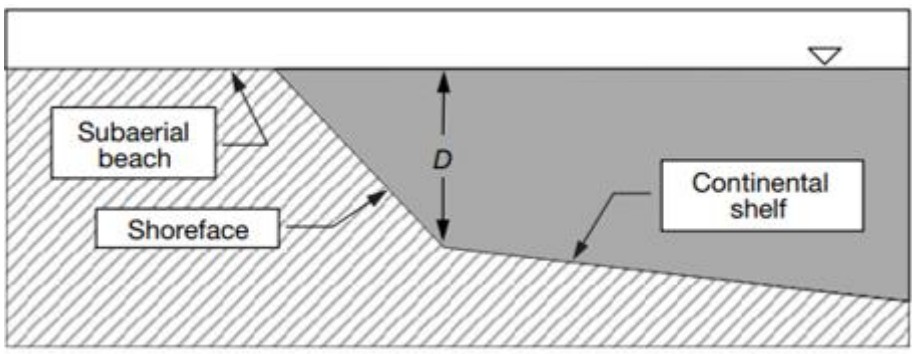

**Figure 2: Cross-sectional profile of CEM showing the location of the depth of closure, where the shoreface slope intersects the continental shelf slope (after Ashton et al., 2001).**

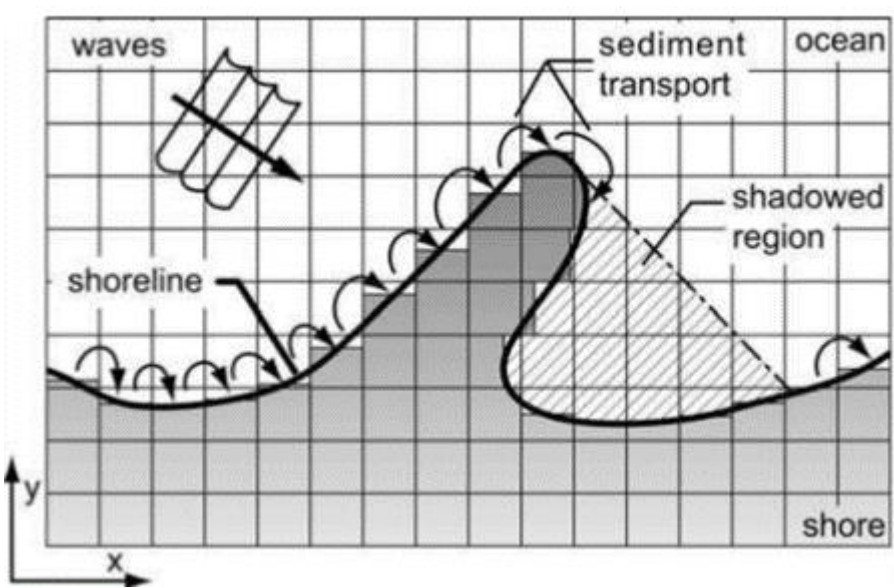

Figure 3: Plan-view schematic of CEM showing the shadow zone that is formed when protruding sections of coastline prevent waves from approaching the shoreline (Ashton and Murray, 2006a).

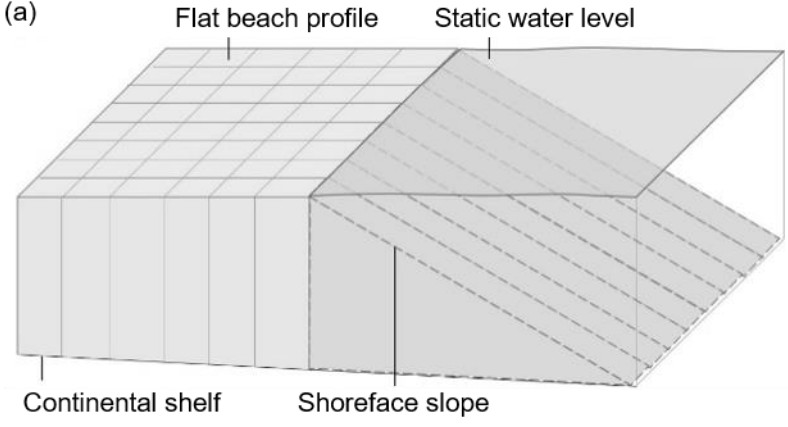

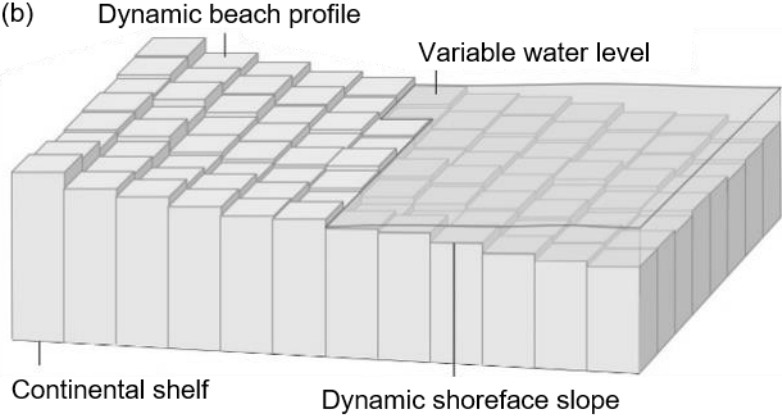

**Figure 4: Schematics of CEM (a) and CEM2D's (b) profiles, illustrating the difference in structure and dimensionality of the two models.**

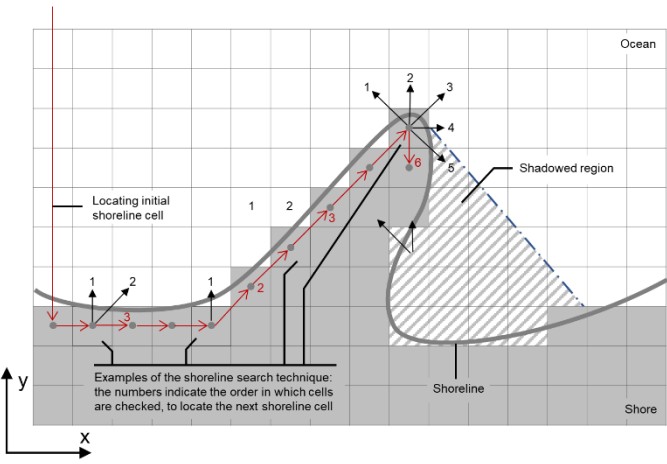

**Figure 5: A schematic of the shoreline search technique used in CEM (and CEM2D) to map the X and Y location of the shoreline cells. The number in square brackets denotes the shoreline cell number that is associated with a particular X and Y value and the number on each arrow is the iteration of the clockwise search from the shoreline cell where it originates.**

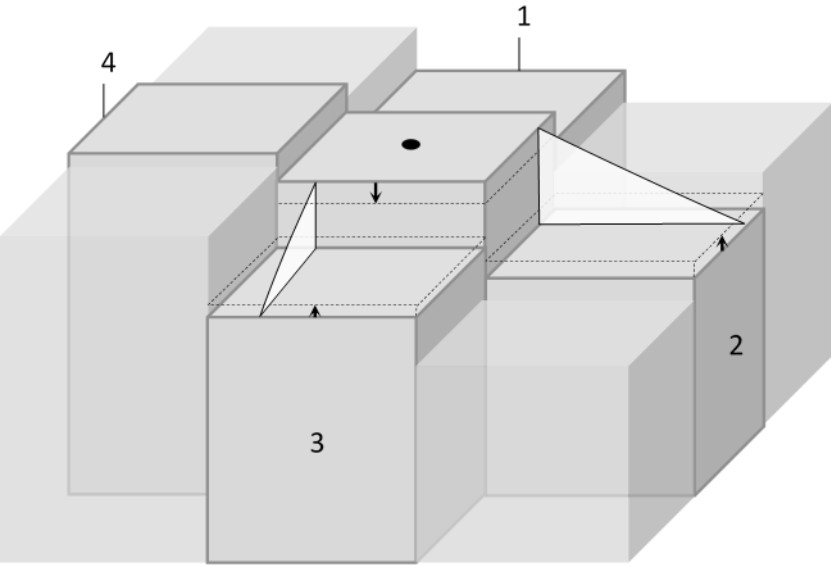

Figure 6: Schematic of the sediment distribution technique used to distribute sediment to cells with lower elevations. In the example, the angle between the central cell and cells [2] and [3] exceeds the threshold for diffusion. Sediment is removed from the central cell and redistributed to these cells. Cells [1] and [4] are not readjusted in this iteration but may be in subsequent sweeps of the coastline.



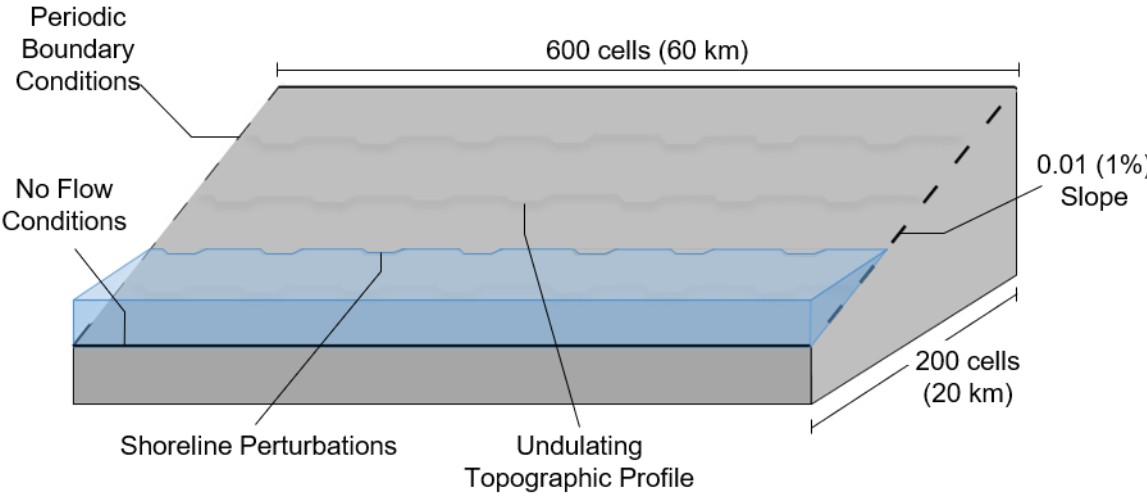

**Figure 7: A schematic of CEM2D's model set-up and initial conditions used for simulations presented in this paper.**

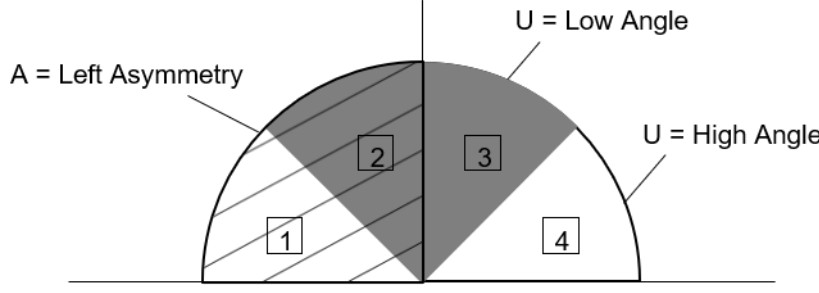


**Figure 8: Schematic showing the wave angle direction, defined by the wave climate asymmetry (A) and the proportion of high to low angle waves (U) with the numbers denoting the four bins.**

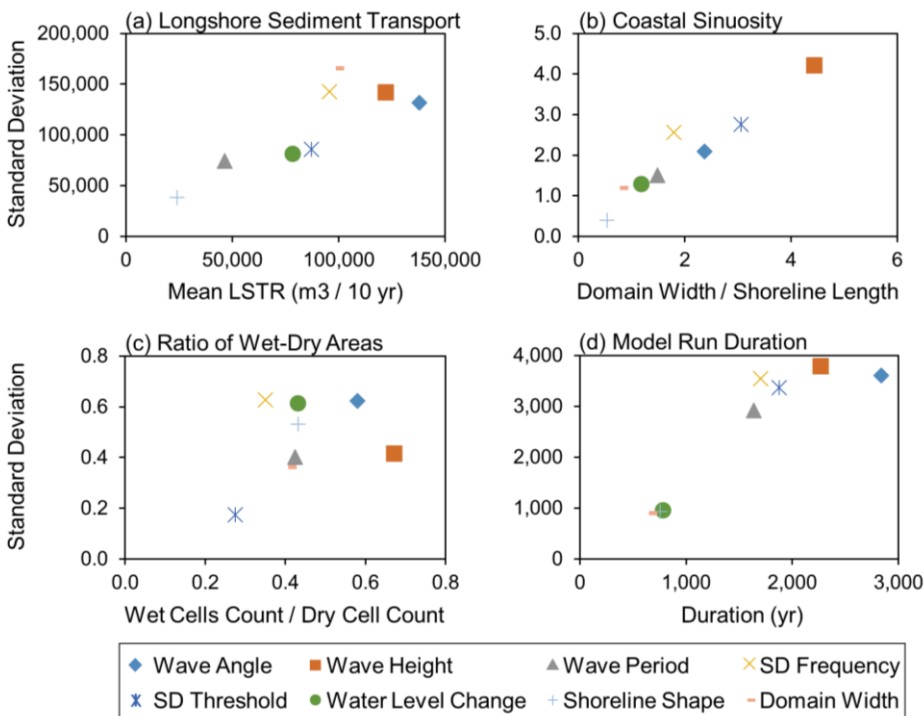

**Figure 9: The mean and standard deviation of results from the input factors, according to the four behavioural indices labelled a-d.**

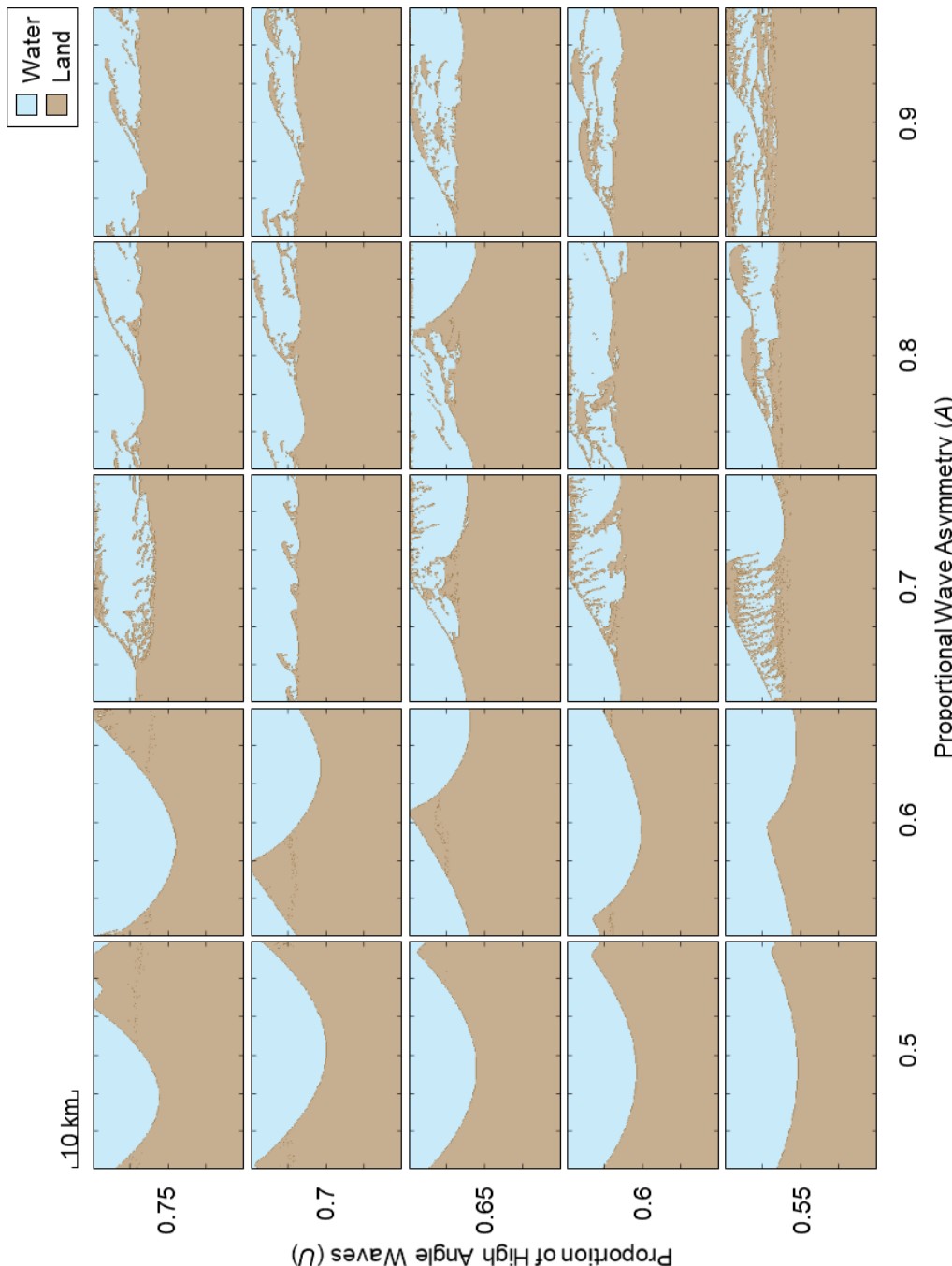

**Figure 10: A matrix of results from CEM showing final shoreline morphologies as a function of the wave angle asymmetry (A) and proportion of high angle waves (U) approaching the coast relative to the local shoreline orientation. The outputs measure 20 km width and 30 km in length and are not inclusive of the periodic boundaries.**


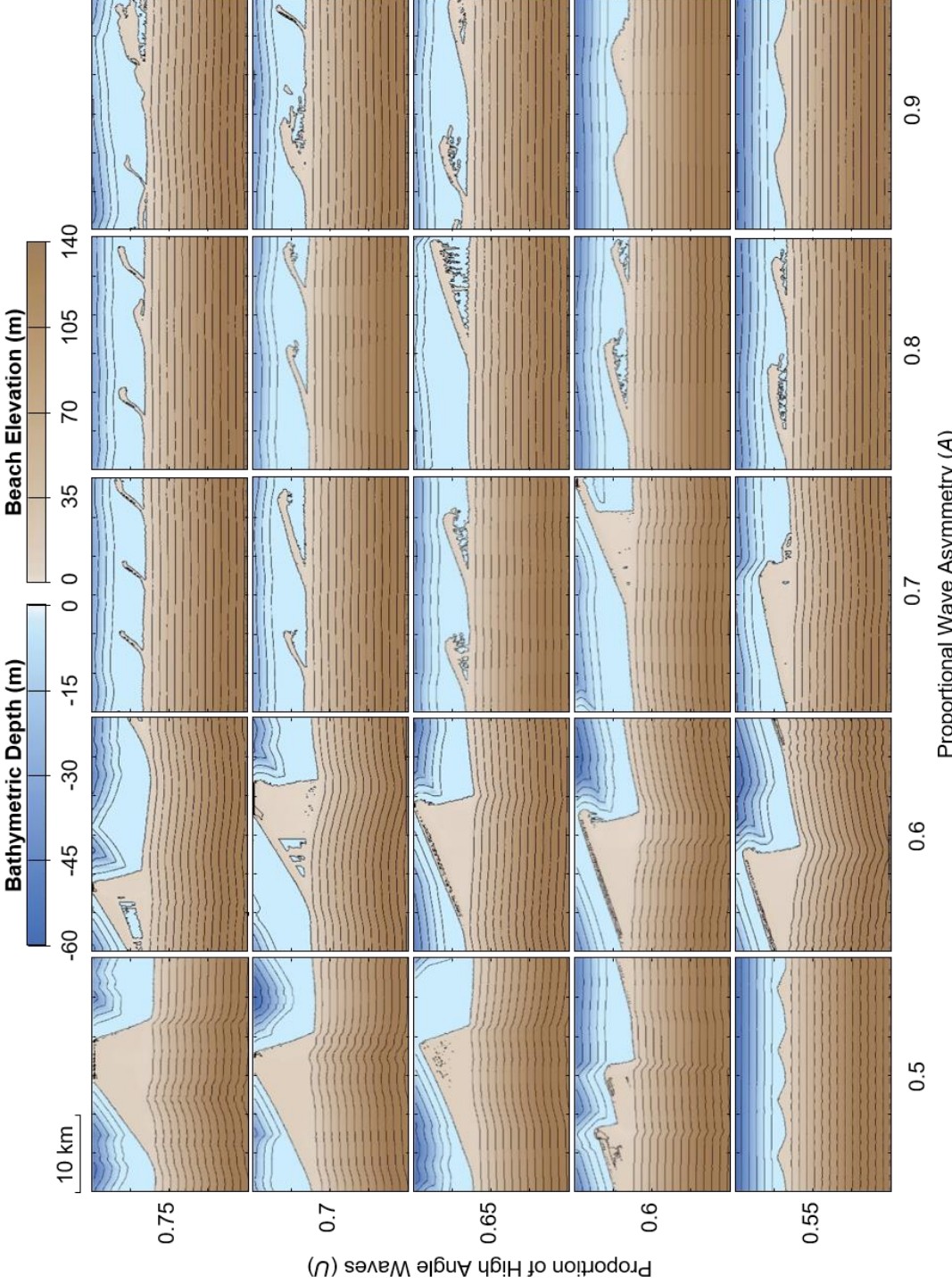

**Figure 11: A matrix of results from CEM2D showing two-dimensional final shoreline morphologies as a function of the wave angle asymmetry (A) and proportion of high angle waves (U) approaching the coast relative to the local shoreline orientation. The outputs measure 20 km width and 30 km in length and are not inclusive of the periodic boundaries.**

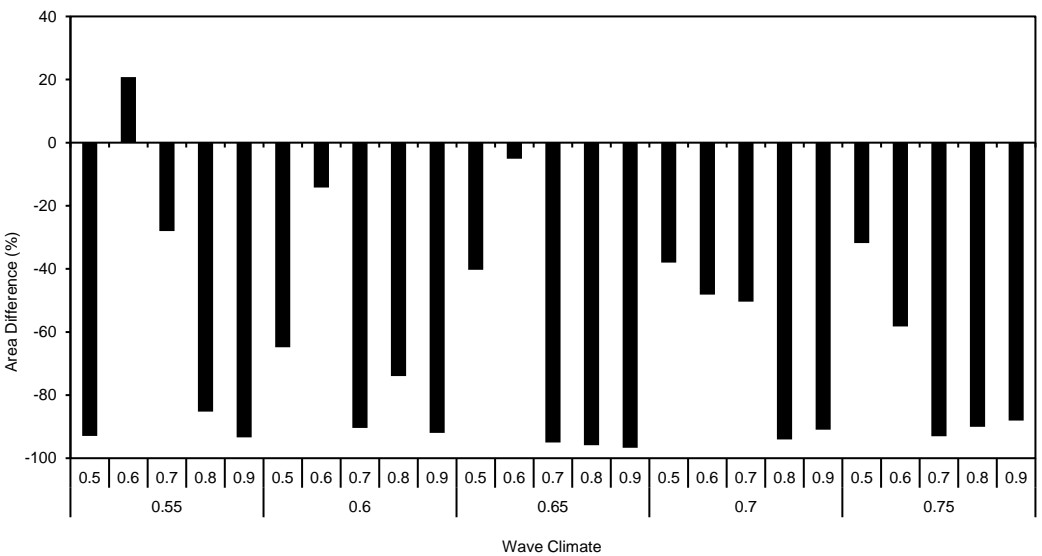


**Figure 12: Areal difference (%) between results of CEM2D to CEM. The wave climate given along the x-axis is defined according to the wave asymmetry (top row) and the proportion of high angle waves (bottom row).**

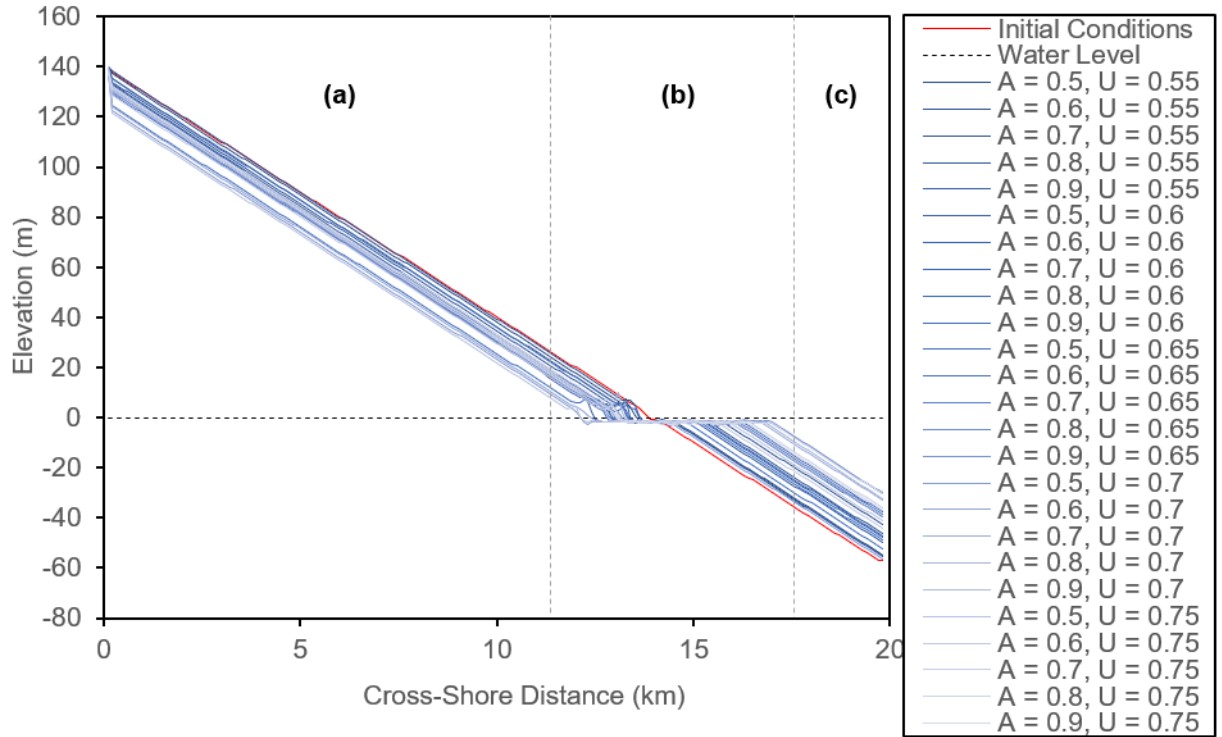


**Figure 13:** Cross-shore profiles taken for each of the twenty-five simulations, with water level shown as a dashed line and the initial cross-shore profile as a solid red line. Labelled are the (a) beach surface, (b) dynamic shoreline and upper nearshore and (c) the lower nearshore.

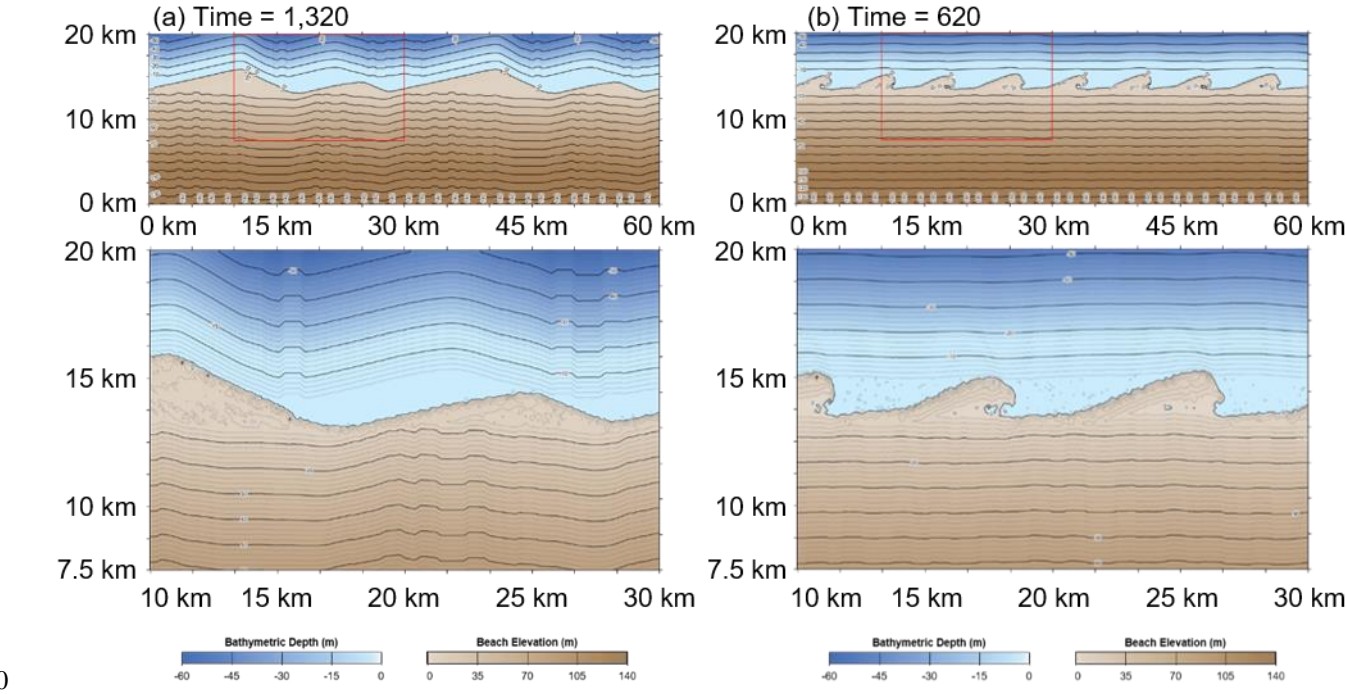


**Figure 14: Morphology plots showing outputs of (a) A = 0.6, U = 0.6 at Time = 1,320 and (b) A = 0.7, U = 0.65 at Time = 620.**

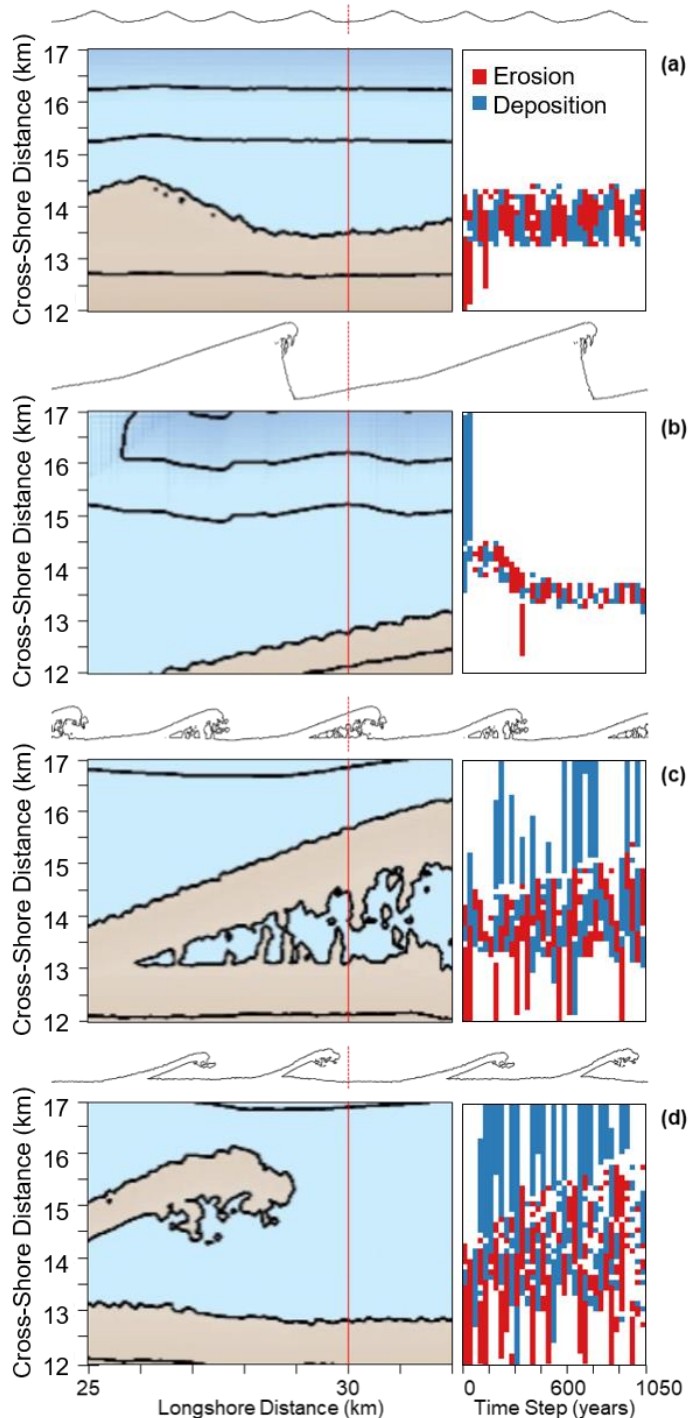

**Figure 15: Shoreline Morphologies (left) and volume stacks (right) for four simulations where the wave climate is defined by (a) A = 0.5, U = 0.55, (b) A = 0.6, U = 0.6, (c) A = 0.7, U = 0.65 and (d) A = 0.8, U = 0.7. The red line marks the cross-shore transect where the change in volume at 30 year time intervals is recorded.**

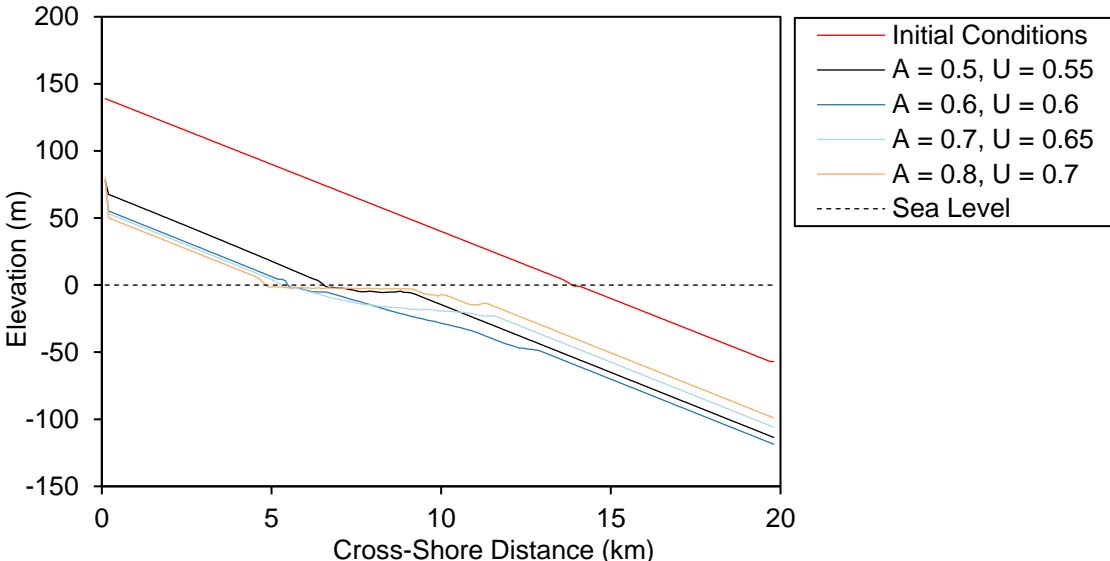

**Figure 16: Cross-shore coastal profiles for four simulations where the wave climate is defined by (a) A = 0.5, U = 0.55, (b) A = 0.6, U = 0.6, (c) A = 0.7, U = 0.65 and (d) A = 0.8, U = 0.7. The initial profile is given as a solid red line and the water level as a dashed black line.**

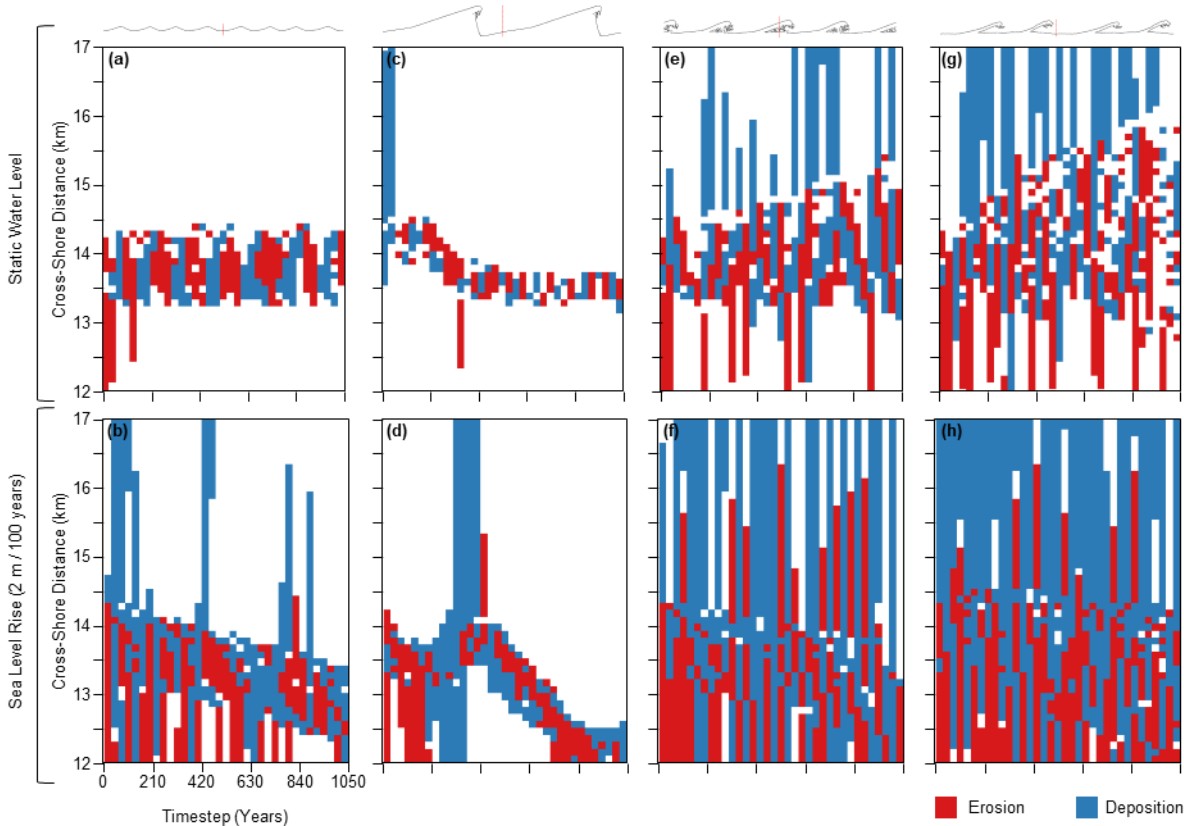

**Figure 17: Volume stacks for four wave climate conditions defined by (a-b) A = 0.5, U = 0.55, (c-d) A = 0.6, U = 0.6, (e-f) A = 0.7, U = 0.65 and (g-h) A = 0.8, U = 0.7. Results with a static water level are shown along the top row (a, c, e, g) and with sea level rise at a rate of / 100 years along the bottom row (b, d, f, h). The shoreline outlines at the top of the figure are taken from the static water level scenarios and the red line marks the cross-shore transect where the change in volume at 30 year time intervals is recorded.**


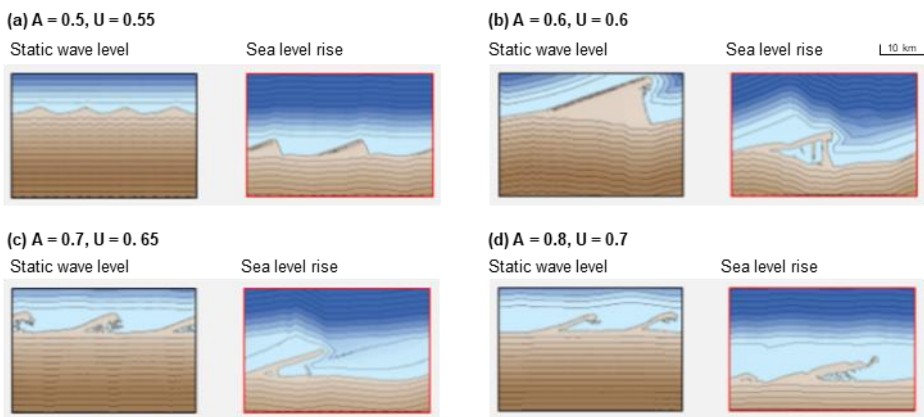

**Figure 18: Coastal morphologies for four simulations where the wave climate is defined by (a) A = 0.5, U = 0.55, (b) A = 0.6, U = 0.6, (c) A = 0.7, U = 0.65 and (d) A = 0.8, U = 0.7, run with two water level scenarios including a static water level (left, black outline) and a rising sea level at a rate of 2 m / 100 years (right, red outline).**