# Peer review of "The Coastline Evolution Model 2D (CEM2D) V1.1"

_Geoscientific Model Development, 2019_

## Referee Comment (RC1) · Eli D Lazarus (Referee) · 15 Oct 2019

**GMD-2019-197 REVIEW**

In this manuscript, the authors introduce a 2DH version of the Coastal Evolution Model (CEM) that is able to account for, among other variables, a dynamic sea level over extended spatial scales (tens to hundreds of kilometres) and "meso" time scales (decades to centuries).

I am familiar with the original (proto) CEM, based on Ashton *et al.* (2001) and its conversion since to an open-source landscape-evolution model at the core of undertakings like the Community Surface Dynamics Modelling System (CSDMS).

Overall, I found this manuscript readable and digestible – which is excellent for a model description. The Introduction grounds the model in a helpful context, and provides the tool with its *raison d'être*.

I have one major comment and a few minor ones – remarks that I hope will improve the manuscript that much more.

**Major comment**

I ended up confused about how the "variable water level" advance gets presented here. At L111, the authors write that "CEM2D contains a significant number of modifications to enable it to model the evolution of coastal features including their topographic profiles and to study the influence of a variable water level," but then at L179 state that "we do not examine the influence of a variable water level on coastal morphodynamics but explore the changes 180 that happen with a two-dimensional evolution of the coastal profile."

What wasn't absolutely clear to me is what we gain atop what CEM will already demonstrate. I would encourage the authors to show, as explicitly as they can, the difference in results yielded by CEM vs CEM2D. I see Figs. 10 & 11 – I mean some kind of quantitative demonstration of the differences in output?

I think part of what I don't understand is why the "new" parts of CEM2D are underplayed here. Most of this paper comes across as a kind of reassurance that the new version still does everything the old version does. I appreciate the addition of elevation contours in Fig. 11 (and others), but I finished the manuscript still waiting for the other shoe to drop. Here's what the old model did; and here's the amazing thing this one does. I wanted a clearer demonstration of the latter. This manuscript seems like exactly the opportunity to showcase everything that the inclusion of a variable water level now allows – instead of a conservative assurance that there's been no loss of benefits from CEM.

In CEM, is it possible to impose a linear erosion rate to simulate sea-level rise? And then, in CEM2D, could the authors show the equivalent experiment with the addition of an actual landscape gradient? Seems to me that, given its emphasis on variable water level as the key motivation for CEM2D, this paper needs to focus on what that yields. I think the authors are starting to get there in Section 5.3 (L287) and at L323 in the Discussion, but we need clearer and more specific supporting evidence than Fig. 14 provides.

**Minor comments**

L47 – "specific research questions" – Theoretical explorations can also derive from "specific" research questions. Do you mean questions that pertain to spatially explicit problems? (Simulating a particular reach of coastline?) Suggest rethinking this paraphrasing of Murray (2007).

L323 – clarify sentence – (see "systems", plural?)

L335 – Confusing paragraph (and see punctuation of "waves") – I think the authors are trying to convey what was absent from CEM but is now present in CEM2D, but the paragraph doesn't read that way to me. Sounds like it's all still "to be calculated", as though these capacities don't yet exist in CEM2D. (But they do, correct?)

Fig. 2 – more labels? alongshore/crossshore, etc., to match Fig. 4?

Fig. 9a, d – x axes? It's not clear to me what's plotted here.

Fig. 10 – Labels for land versus sea? Strange visual inversion when sideways (Fig 11 clearer).

Fig. 12 – White band? I'm missing it, I think?

I wish the authors a fruitful revision, and look forward to seeing this work in print.

– *Eli Lazarus*

E.D.Lazarus@soton.ac.uk

---

## Author Comment (AC1) · 1 Nov 2019

Thank you for your comments, Eli. We have taken your feedback on board and would like to give a brief response before all reviews on the manuscript have been received.

The paper was submitted as a development and technical paper and therefore, is deliberately light on the results. However, we fully appreciate the need to provide a greater demonstration of what the new CEM2D is capable of and we will look to make changes in the paper to this effect.

Thank you again for your comments, Chloe

---

## Referee Comment (RC2) · Anonymous Referee #2 · 18 Dec 2019

The work presented in this paper extends the Coastal Evolution Model (CEM, Ashton and Murray, 2006a) including a two-dimensional sediment redistribution numerical scheme based on a slope stability condition. As in CEM, the model responds to a single process, the alongshore gradient in sediment transport, allowing for the redistribution of sediments along the coast. In addition, the geometric condition on the slope stability allows for the numerical redistribution of sediment in adjacent cells. The authors unravel the sensitivities, or the signal-noise ratio, that several factors originate on a few behavioral indices output by the model. The new CEM 2D model is compared with the original CEM through a numerical experiment. In this numerical experiment, both models are loaded with wave climates accounting for several directional distribution of waves with the same height and period in order to explore the morphological features originated in plan-view by both models in the very long term (3000yr).

**Major comments**

The authors claim two novelties in CEM2D in comparison with CEM. The redistribution of sediment in the cross-shore direction and the response to changes in sea level. However, in the present document, the improvements that this numerical redistribution of sediments introduces in landscape evolution over the previous model are not clear. And the response to changes in sea level is even not explored through the paper. I have also got the impression comparing Figure 10 and Figure 11 that this redistribution changes the spatial scale of the features developed due to probably a reduction on the effective alongshore flux due to this sediment redistribution scheme. The CEM2D model produces very different results for example for A=0.5 with any U and what seems to be a numerical instability for U=0.75, A=0.6. The authors should have examined these big differences in model behavior. In general, it is important to highlight what new features are better represented in the new version and what are the implications for coastline shape simulation, and a simple comparison with the old model results is not enough. For example, Antolinez et al. (2018) use CEM for hindcasting 150 of shoreline evolution in the Carolinas capes, but they don't account for changes in sea level, this new CEM2D model brings a great opportunity to account for this process adequately and to show what CEM is missing.

The authors claim this new model is a 2D model, however the model is still a single process model, alongshore sediment transport, with numerical diffusion in 2D using a stability slope condition. The model would not work in waves perpendicular to the coast. In my opinion, it is not consistent to claim a 2D model that is only solving alongshore sediment transport with an integrated semi-empirical formula, why the authors don't solve sediment transport at cell level? How is this integrated transport redistributed in the cross-shore? Or is it all taken from the adjacent cell to the shoreline position? If the last, I have the impression that the model would create spurious shoreline change behavior as it has the possibility to remove a lot of sand from the adjacent cell to the shoreline in alongshore direction and later on the need to redistribute sediment in the cross-shore direction due to the slope criteria, when in nature sediment would have been taken gradually from several cells in cross-shore direction; your slope condition is changing the cross-shore profile shape in the upper-shoreface in time, could you validate this?

I also miss a lot of discussion and review of recent existing models accounting for alongshore and cross-shore responses and accounting for changes in sea level, for example, Larson et al. (2016), Vitousek et al. (2017), Robinet et al. (2018), and Antolinez et al. (2019).

In the abstract the authors explain the model is suitable for evolving morphological features in time scales from 10 to 100 years, but any analysis is performed in these timescales.

I support the idea of changing the bathymetry, but what is the added value if wave transformations are still assuming parallel contours to the shoreline as in CEM? other models such as Robinet et al., 2018 already account for a scheme propagating waves in complex bathymetry and studies such as the one presented in Limber et al. (2017) proofs its importance.

I can read several times through the text the authors acknowledge certain model limitations and they propose to incorporate improvements in coming versions, why do not incorporate them now? (for example, lines 238-240)

Certain Figures are not properly presented, for example Figure 10 and Figure 11 cut the model domain and shoreline shapes are not complete, Figure 9 has different color markers in the legend than in the subplots.

I encourage the authors to review their analysis and resubmit a complete new version making emphasis in validating what new features their 2D numerical scheme brings in landscape evolution and encourage them to discuss the main points I have made through the text (e.g. profile shape, numerical sediment diffusion, integrated vs in cell sediment transport computations, sea level changes, wave propagation in complex bathymetry, cross-shore processes, …).

**References**:

Antolínez, J. A. A., Murray, A. B., Méndez, F. J., Moore, L. J., Farley, G., & Wood, J. (2018). Downscaling changing coastlines in a changing climate: The hybrid approach. *Journal of Geophysical Research: Earth Surface*, *123*, 229–251. https://doi.org/10.1002/2017JF004367

Antolínez, J. A. A., Méndez, F. J., Anderson, D., Ruggiero, P., & Kaminsky, G. M. (2019). Predicting climate-driven coastlines with a simple and efficient multiscale model. *Journal of Geophysical Research: Earth Surface*, *124*. https://doi.org/10.1029/2018JF004790

Larson,M., Palalane, J., Fredriksson, C., & Hanson, H. (2016). Simulating cross-shore material exchange at decadal scale. theory and model component validation. *Coastal Engineering*, *116*, 57–66.

Limber, P. W., Adams, P. N., & Murray, A. B. (2017). Modeling large-scale shoreline change caused by complex bathymetry in low-angle wave climates. *Marine Geology*, *383*, 55–64. https://doi.org/10.1016/j.margeo.2016.11.006

Robinet, A., Idier,D., Castelle, B.,&Marieu, V. (2018). A reduced-complexity shoreline change model combining longshore and cross-shore processes: The lx-shore model. *Environmental Modelling & Software*, *109*, 1–16. https://doi.org/10.1016/j.envsoft.2018.08.010

Vitousek, S., Barnard, P. L., Limber, P., Erikson, L., & Cole, B. (2017). A model integrating longshore and cross-shore processes for predicting long-term shoreline response to climate change. *Journal of Geophysical Research: Earth Surface*, *122*, 782–806. https://doi.org/10.1002/2016JF004065

---

## Author Comment (AC2) · 26 Jan 2020

**Author's Response to Reviewer Comments**

**Authors comments in bold.**

**We would like to thank both reviewers for their constructive comments on this manuscript. We would like to address each of your comments and the changes we have made to the paper in response to your feedback.**

**Due to the addition of text and subsections, some of the page and section numbers given by the reviewers may now have change. Page and section numbers in the responses given by the author are for the track changed document, whereby 'No Markup' is shown.**

**Comments from the editors and reviewers:**

**Reviewer 1: Minor Comments**

| 1 | L47 – "specific research questions" – Theoretical explorations can also derive from "specific" research questions. Do you mean questions that pertain to spatially explicit problems? (Simulating a particular reach of coastline?) Suggest rethinking this paraphrasing of Murray (2007). |
|---|---|
| | **L47 changed to:**
 *"rather than pertaining to spatially explicit research questions"* |

| 2 | L323 – clarify sentence – (see "systems", plural?) |
|---|---|
| | **L376 changed to:**
 *"profile of the coastal system changes"* |

| 3 | L335 – Confusing paragraph (and see punctuation of "waves") – I think the authors are trying to convey what was absent from CEM but is now present in CEM2D, but the paragraph doesn't read that way to me. Sounds like it's all still "to be calculated", as though these capacities don't yet exist in CEM2D. (But they do, correct?) |
|---|---|
| | **The sediment transport equations in CEM2D do not currently take into consideration the water depth, or how far from the shore the waves break. The equations used are inherited from the original CEM and these have not been updated in this version of the model. Subsequent versions of the model will include revisions of the sediment transport equations so that these variables are taken into account, but this is a significant change that will require a substantial amount of work and so has been reserved for the next iteration of the model.**

**L386 has been changed to make this point clearer:**
 *"The longshore sediment transport equations in CEM2D are inherited from the CEM and currently do not take into consideration the water depth, or how far from the shore the waves break. Since the water depth can now be calculated within this 2D model, future developments of CEM2D will focus on a revision of the sediment transport equation and include a more suitable calculation that can take advantage of the increased complexity and added functionalities in CEM2D."* |

| 4 | Fig. 2 – more labels? alongshore/crossshore, etc., to match Fig. 4? |
|---|---|
| | **Figure 2 has been updated to include more labels and to show the shoreline search technique more clearly.** |

| 5 | Fig. 9a, d – x axes? It's not clear to me what's plotted here. |
|---|---|
| | **Figure 9 has been updated to make the axis labels clearer.** |

| 6 | Fig. 10 – Labels for land versus sea? Strange visual inversion when sideways (Fig 11 clearer) |
|---|---|
| | **Figure 10 has been updated to include 'land' and 'water' labels.** |

| 7 | Fig. 12 – White band? I'm missing it, I think? |
|---|---|
| | **Figure 12's caption has been updated to:**
 *"Cross-shore profiles taken for each of the twenty-five simulations, with water level shown as a dashed line and the initial cross-shore profile as a solid red line. Labelled are the (a) beach surface, (b) dynamic shoreline and upper nearshore and (c) the lower nearshore."* |

**Reviewer 1: Major Comments**

| 8 | I ended up confused about how the "variable water level" advance gets presented here.

This manuscript seems like exactly the opportunity to showcase everything that the inclusion of a variable water level now allows – instead of a conservative assurance that there's been no loss of benefits from CEM. |
|---|---|
| | **Both of these points highlight that we do not explore the variable water level function to a great depth in this paper. The aim of this paper was to outline the technical developments of CEM2D and the advantages of these (e.g. dynamic bathymetry and variable water level), whilst retaining the original model's ability to simulate fundamental cause-effect relationships which has received much credibility.**

**We have carried out significant novel research using the variable water level function to look at sea level rise impacts and as such, we felt it was better placed in the follow up results paper which is currently being finalsed. We do however recognise that it could be beneficial to add some additional detail into this paper about the variable water level function, given that it is one of the major developments. In light of this, the following changes have been made:**

**A significant additional section (5.5 Variable Water Level) has been added to the results section:**
 *"5.5 Variable Water Level*
 *Changing the water level against the dynamic topography allows CEM2D to explore how a rising water level might affect how coastal systems behave. The results demonstrate that a rising sea level causes landward recession of the shoreline and uplift of the profile (Figure 15), as is commonly held (Dickson et al., 2007; Bird, 2011). The rate of recession is broadly within two orders of magnitude the rate of sea level rise, prescribed at 2 m / 100 years in the simulations, which is in agreement with Bruun Rule estimations (Bruun, 1962). Variations in the rate of recession and morphology of the cross-shore profile are, however, observed with* |

*different wave climate conditions that differ in the balance of cross-shore and longshore flows (Figure 15).*

*As in Figure 14, Figure 16 shows the change in volume of sediment across a transect (x = 30 km) every 30 simulated years, for four wave climate scenarios where (a-b) A = 0.5, U = 0.55, (c-d) A = 0.6, U = 0.6, (e-f) A = 0.7, U = 0.65 and (g-h) A = 0.8, U = 0.7. The figure shows a comparison of the results with a static water level (top) and with a rate of sea level rise of 2 m / 100 years (bottom). The spatial extend of morphological change is more diverse and widespread when the systems are subject to sea level rise. The principal active zone also tracks backwards as the water level rises and the shoreline recedes.*

*A rising sea level influences the evolution of shoreline features that evolve in the model, including cusps (a), sand waves (b), reconnecting spits (c) and flying spits (d) (Figure 17). As also shown in Figure 17, recession of the shoreline is observed in all four coastal systems as the water level rises regardless of the wave climate conditions (also shown in Figure 15). Where the wave climate is symmetrical, cuspate features form under a static water level but have a slight asymmetry under sea level rise conditions, where the direction bias in the model is exaggerated. The cusps extend further offshore where the bays between the headlands are eroded, increasing wave shadowing and hence, exaggerating the effects of the directional bias. A slight asymmetry in the wave climate forms sand waves along the shoreline (Figure 17b), but submergence of these features under a rising sea level leads to the formation of a lagoon in the low-lying centre of the landform. Where the wave climate is defined by A = 0.7, U =0.65 (Figure 17c), reconnecting spits form when the water level is static, but as the water level rises the pathways that reconnect the spit to the mainland are submerged. In Figure 17d, flying spits are shown to evolve with and without sea level rise, where the wave climate is highly symmetric. The difference between these two simulations is that under a rising water level, the flying spits cycle through submergence and reformation, as they are drowned by the rising water, but new features are able to form due to the high rate of longshore sediment transport. Remnants of the submerged spits remain in the nearshore and promote the development of spits in these areas due to the shallower water and also influence the unique plan-form morphology of the features."*

Accordingly, the text has also been edited in Section 6 Discussion from L402:
*"A key component of CEM2D is its variable water level. If we are to explore coastal evolution over the mesoscale, being able to model the effect of rising sea levels is essential. Whilst we have not exhausted the uses of this function here, we have demonstrated its development and how it is facilitated in the model. The power of this tool is vast and will be particularly useful for coastal managers who must plan for the dynamic evolution of these system over time periods that will be highly influenced by the effects of climate change. The results presented show that a rising sea level can significantly influence the evolution of coastal systems through recession and uplift of the cross-shore profile (Bruun, 1962), the types of shoreline features that form and the way in which these features evolve. Also found was the clear role of the wave climate conditions on the morphology of the shoreline and landform features behaviour, as the water level rises."*

Furthermore, we have added the following text to Section 4.3, clarifying the scope of the paper:
*"The primary purpose of this paper is to highlight the technical development of CEM2D and demonstrate its additional functionalities. The simulations shown focus on how the coastal*

*systems evolve with an unchanging water level at 0 m elevation, but results are also given for how an increasing water level at a rate of 2 m / 100 years influences the evolution of four shoreline types: cuspate, sand wave, reconnecting spit and flying spit. This rate of rise is in line with the UK Climate Projections 2009 (UKCP09) (Jenkins et al., 2009) H++ scenario of 0.93-1.9 m sea level rise by 2100 (Jenkins et al., 2009; Lowe et al., 2009)."*

| 9 | I would encourage the authors to show, as explicitly as they can, the difference in results yielded by CEM vs CEM2D. I see Figs. 10 & 11 – I mean some kind of quantitative demonstration of the differences in output? |
|---|---|
|   | Here's what the old model did; and here's the amazing thing this one does. I wanted a clearer demonstration of the latter. |

**There are two primary differences between CEM and CEM2D. This first is in the representation of the coastal domain, with CEM2D having a dynamic topography and bathymetry that evolves through the exchange of sediment between cells. The second development is in the ability of CEM2D to impose a variable water level in the coastal system. We have added some additional results to explore these functions more explicitly, as detailed below.**

**A significant additional section has been added to show some quantitative results regarding the differences in spatial scale of the features: 5.3 Spatial Scale of Shoreline Features. Within this section, it is highlighted that the differences in spatial scale of features between CEM and CEM2D are likely to occur due to the different way in which the domain is representation and sediment is distributed; as stated above, this is one of the primary developments of CEM2D. The text below is taken from this additional subsection:**

*"5.3 Spatial Scale of Shoreline Features*
*The spatial scale of shoreline features differs between results from CEM and CEM2D. Metrics from the end of each run, as shown in Figures 10 and 11, show larger features evolve in CEM2D in six of the simulations. The larger features evolve under wave climate conditions where A = 0.6, U = 0.55-0.65 (sand waves), where A = 0.7-0.8, U = 0.7 (flying spit) and where A = 0.9, U = 0.75 (flying spit) and smaller features evolve in the remaining nineteen simulations. However, each run terminates at a different timestep and a comparison of results at the earliest termination for each pair of simulations shows that in all but one of the runs (A = 0.6, U = 0.55), the features are smaller and less developed in CEM2D than the CEM (Figure 18).*

*The evolution of landforms is more gradual in CEM2D, likely as a result of differences in the representation of the domain and in the distribution of sediment. Rather than sediment being distributed evenly across the nearshore to the depth of closure, as in CEM, CEM2D uses the sediment distribution method to route sediment along lines of steepest descent and spreads available material across the nearshore profile. This leads to both the formation of shoreline features but also to the formation of a shallow nearshore shelf (see Section 5.4).*

*Highlighted above are differences in results between CEM and CEM2D and in particular, the complexity of results generated in CEM2D due to the addition of a dynamically evolving profile. In nature, the features discussed evolve at different rates and to different spatial scales depending and in order to use CEM2D to investigate such systems, parameters in the model*

*including the threshold and frequency of sediment distribution should be adjusted to suit the specific environment studied and the rates at which these features form."*

A significant additional section has been added to show some quantitative results regarding the influence that a rising water level has on the coastal systems: 5.5 Variable Water Level. The principal results are as follows (see response number 8 for full text):

L358 *"The results demonstrate that a rising sea level causes landward recession of the shoreline and uplift of the profile (Figure 15), as is commonly held (Dickson et al., 2007; Bird, 2011). The rate of recession is broadly within two orders of magnitude the rate of sea level rise, prescribed at 2 m / 100 years in the simulations"*

L361 *"Variations in the rate of recession and morphology of the cross-shore profile are, however, observed with different wave climate conditions"*

L368 *"The spatial extend of morphological change is more diverse and widespread when the systems are subject to sea level rise. The principal active zone also tracks backwards as the water level rises and the shoreline recedes."*

L371 *"A rising sea level influences the evolution of shoreline features that evolve in the model, including cusps (a), sand waves (b), reconnecting spits (c) and flying spits (d) (Figure 17)."*

The text has also been edited accordingly in Section 6 Discussion from L402, as detailed in response number 8.

| 10 | In CEM, is it possible to impose a linear erosion rate to simulate sea-level rise? And then, in CEM2D, could the authors show the equivalent experiment with the addition of an actual landscape gradient? … we need clearer and more specific supporting evidence than Fig. 14 provides. |
|---|---|
| | We're trying to move away from the idea of imposing a linear erosion rate to simulate sea level rise, but rather in CEM2D we can increase the water level and investigate how it affects the existing erosion rate – thereby just changing sea level – rather than having to adjust the erosion rate. |

We acknowledge that we need more evidence to show the variable water level function and have added results from this research into subsection 5.5 Variable Water Level. The full text from this additional section is given in response number 8 and the key findings are given below (also given in response number 9):

L358 *"The results demonstrate that a rising sea level causes landward recession of the shoreline and uplift of the profile (Figure 15), as is commonly held (Dickson et al., 2007; Bird, 2011). The rate of recession is broadly within two orders of magnitude the rate of sea level rise, prescribed at 2 m / 100 years in the simulations"*

L361 *"Variations in the rate of recession and morphology of the cross-shore profile are, however, observed with different wave climate conditions"*

**L368** *"The spatial extend of morphological change is more diverse and widespread when the systems are subject to sea level rise. The principal active zone also tracks backwards as the water level rises and the shoreline recedes."*

**L371** *"A rising sea level influences the evolution of shoreline features that evolve in the model, including cusps (a), sand waves (b), reconnecting spits (c) and flying spits (d) (Figure 17)."*

**The text has also been edited accordingly in Section 6 Discussion from L402, as detailed in response number 8.**

**Reviewer 2: Major Comments**

| 11 | The improvements that this numerical redistribution of sediments introduces in landscape evolution over the previous model are not clear. |
|---|---|

**We are sorry if this is not clear. In CEM the horizontal filling and emptying of cells can allow too much sediment to be added or removed from cells which then translates into the simplistic landward or seaward movement of the shoreline. This method is rather opaque in existing descriptions of the original CEM model and only really clear when analyzing the code. In CEM2D and in this description we have been very clear about the need to refine this process, especially in light of the model moving towards being two dimensional and the storage of sediment in cells being vertical as opposed to horizontal.**

**Importantly, the CEM2D sediment distribution method enables the model to simulate the dynamic topography of the coastal system not just the shore edge. Sediment transport is first calculated for the one-line shoreline, like the CEM, but rather than the material assumed to be distributed evenly across the nearshore profile, the sediment distribution method allows the profile to dynamically evolve. The method is based broadly on the sand pile theory and steepest descent method, which are used in other numerical models.**

**To make this clearer in the manuscript, the following text has been added to L140 in Section 3:**
   *"The implementation of this method in CEM2D allows the nearshore profile to evolve dynamically, rather than assuming an even distribution across the nearshore profile and forming shore-parallel contours, as is the case in CEM and other one-line models."*

**The differences observed between the CEM2D and CEM due to the implementation of this method are highlighted in Section 5.3 Spatial Scale of Shoreline Features, which has been added to the manuscript. An extract from L302 is shown below (see full text in response number 9):**
   *"The evolution of landforms is more gradual in CEM2D, likely as a result of differences in the representation of the domain and in the distribution of sediment. Rather than sediment being distributed evenly across the nearshore to the depth of closure, as in CEM, CEM2D uses the sediment distribution method to route sediment along lines of steepest descent and spreads available material across the nearshore profile. This leads to both the formation of shoreline features but also to the formation of a shallow nearshore shelf (see Section 5.4)."*

| 12 | The response to changes in sea level is even not explored through the paper. |
|---|---|
| | **This feedback was also given by reviewer 1 and changes have been made accordingly, through the addition of Section 5.5 Variable Water Level and additional text in Section 6 Discussion, as detailed in response number 8 above.** |

| 13 | I have also got the impression comparing Figure 10 and Figure 11 that this redistribution changes the spatial scale of the features developed due to probably a reduction on the effective alongshore flux due to this sediment redistribution scheme. |
|---|---|
| | **The spatial scale of features is different between CEM and CEM2D, due to the way in which sediment is distributed in both models. This has now been addressed in the manuscript, with the addition of Section 5.3 Spatial Scale of Shoreline Features (see response number 9 for full text).** |

| 14 | The CEM2D model produces very different results for example for A=0.5 with any U and what seems to be a numerical instability for U=0.75, A=0.6. The authors should have examined these big differences in model behavior. |
|---|---|
| | **Having spent considerable time testing the model we are confident this is not numerical instability. Where A = 0.5, directional bias is observed in the model results due to directional bias found in some of the model routines. The effect is more apparent in CEM2D due to the more complex representation of the domain and sediment distribution in two directions but can also be seen in CEM. This has been highlighted in the manuscript, at L239.** |
| | **For U=0.75 and A=0.6 the instabilities the reviewer may be referring to are ponds/pooling in the CEM2D outputs. This is a reflection and good example of how the 2D representation of a coast in CEM2D allows more complex topography (that evolves over time) to be simulated, rather than the 'single line' of the 1D CEM model that just identifies a shoreline. In our simulations this is not reflective of instability but the cumulative development of the coastline over time.** |

| 15 | In general, it is important to highlight what new features are better represented in the new version and what are the implications for coastline shape simulation, and a simple comparison with the old model results is not enough. |
|---|---|
| | **In our revised manuscript, we identify in several places the new features of CEM2D and what they allow the model to do as progress from CEM. These have been listed below:** |
| | **a. L113** *"CEM2D contains a significant number of modifications to enable it to model the evolution of coastal features including their topographic profiles and to study the influence of a variable water level."* |
| | **b. L115:** *"However, as opposed to each cell containing a fractional horizontal fill of sediment, each cell contains values for depth of sediment to the continental shelf, elevation of sediment above the water level or depth of water (Fig. 5(b)). Having these additional values enables CEM2D to represent two-dimensional coastlines with greater topographic detail compared to the original CEM, as illustrated in Fig. 5."* |

**c. L116:** *"Importantly, the two-dimensional profile allows the morphology of the beach and shoreface to evolve according to the transport of sediment, across the entire model domain. It explicitly models the slope of the continental shelf and shoreface and the morphological profile of the beach and sea floor."*

**d. L126** *"Sediment transport is calculated using the same equation as CEM (Eq. 1) but because CEM2D represents sediment transport in two-dimensions, material eroded from a cell is distributed to the surrounding cells based on the slope between cells and an angle of repose (Fig. 6)."*
…**L146** *"The implementation of this method in CEM2D allows the nearshore profile to evolve dynamically, rather than assuming an even distribution across the nearshore profile and forming shore-parallel contours, as is the case in CEM and other one-line models. The ability of the simulated coast to evolve dynamically in this way provides a more realistic representation of the morphodynamic behaviour of these systems. How sediment is distributed can affect the longer-term evolution of the system and record a morphological memory of landforms which can interact with other features as they form and mature (Thomas et al., 2016)."*

**e. L148** *"CEM2D's two-dimensional structure allows the water level to be varied, but by default the water level is at 0 m elevation. There are two dynamic water level modes within the model which can be run independently or in combination that can be used to represent tidal fluctuations and long-term sea level change."*

**f. L150** *"The increased complexity of the model domain and of sediment transport processes in CEM2D compared to CEM, enable it to model complex two-dimensional coastal profiles and evolve their morphology. The features allow more complex morphodynamic processes to be explored and to investigate not only the evolution of the one-line shore, but the surrounding beach and shoreface."*

**g. L153** *"The sediment storage and handling technique allows complex landforms and features to develop and leave a morphological memory in the bathymetry as they evolve, which is not possible in the equilibrium profile of the CEM."*

**h. L155** "Sea level change is an important addition to this model from the original CEM, that could be used to explore the response of coastal systems to fluctuating water levels and the influence of fundamental climate change effects such as sea level rise."

[revised manuscript text omitted]

**t. L415** *"Using the added functionalities, we have also shown how CEM2D can be used to explore the two-dimensional behaviour and morphodynamic evolution of coastlines and depositional features, over meso-spatiotemporal scales. From the results shown here, it is apparent that the model will enable us to conduct interesting and insightful investigations to answer research questions including how coastal systems behave under changing environmental conditions and how sea level change might influence their morphodynamic behaviour"*

| | |
|---|---|
| 16 | Antolinez et al. (2018) use CEM for hindcasting 150 of shoreline evolution in the Carolinas capes, but they don't account for changes in sea level, this new CEM2D model brings a great opportunity to account for this process adequately and to show what CEM is missing. |
| | **Yes, this would be a great opportunity. Though as in answer 8 to reviewer 1, our purpose with this paper was to outline the technical developments of CEM2D. A paper detailing results from simulations investigating the influence of a variable water level on coastal evolution is currently in preparation, but we do agree that some results that showcase the variable water level function would benefit the manuscript.**

**An additional section (5.5 Variable Water Level) has been added, detailing some results with a variable water level. The text has also been edited accordingly in section 6 Discussion from L402, as detailed in response number 8.** |

| | |
|---|---|
| 17 | The authors claim this new model is a 2D model, however the model is still a single process model, alongshore sediment transport, with numerical diffusion in 2D using a stability slope condition. The model would not work in waves perpendicular to the coast. In my opinion, it is not consistent to claim a 2D model that is only solving alongshore sediment transport with an integrated semi-empirical formula, why the authors don't solve sediment transport at cell level?

How is this integrated transport redistributed in the cross-shore? Or is it all taken from the adjacent cell to the shoreline position? If the last, I have the impression that the model would create spurious shoreline change behavior as it has the possibility to remove a lot of sand from the adjacent cell to the shoreline in alongshore direction and later on the need to redistribute sediment in the cross-shore direction due to the slope criteria, when in nature sediment would have been taken gradually from several cells in cross-shore direction; your slope condition is changing the cross-shore profile shape in the upper-shoreface in time, could you validate this? |
| | **As authors, we discussed the use of the term 2D and concluded that it was suitable for this model since it distributes sediment longshore and cross-shore and also allows the elevation of cells to change according to the volume of sediment they contain. We can change this to quasi-2D or part 2D if the editor thinks this is appropriate.** |

For the second point, as described in Section 3 of the paper, sediment transport is calculated for each shoreline cells and is moved alongshore accordingly, as in CEM. At a defined frequency, calculated according to when a sufficient amount of sediment has built up in these shoreline cells, the sediment will get redistributed. The frequency and threshold for the sediment distribution method is calculated so that enough sediment is moved to be within the threshold of a given slope angle, but not so much that large volumes of sediment are moved and make the system unstable. As a result, sediment is gradually redistributed from as many cells as meet the threshold criteria on each timestep that the method is employed.

We appreciate this may not be completely clear in the manuscript so have modified Section 3 from L123, to make this clearer:

*"In CEM2D the elevation of each cell relative to the water level is used to classify cells as either wet or dry on each model iteration. The boundary between wet and dry is used to locate the shoreline, using the same shoreline search technique as CEM (Fig. 2). As per CEM, Linear Wave Theory is used to transform the offshore wave climate and the CERC formula to calculate sediment flux between shoreline cells (Equation 1). Longshore sediment transport is calculated using the same equation as CEM (Eq. 1) but is then redistributed from the shoreline cells across the model domain using a sediment distribution method which is induced when the slope angle between cells reaches a critical threshold (Fig. 6). This method is based on the relationship between the properties of coastal material (e.g. sand, gravel) and slope angle as shown by McLean and Kirk (1969). We can assume that in general, coastal profiles will maintain an average slope angle consistent with the grain size of beach material although there are a range of factors that can cause steepening or shallowing (McLean and Kirk, 1969).*

*To carry out the sediment redistribution procedure, at user-defined time-step intervals an algorithm sweeps the entire model domain and identifies where a given threshold angle has been reached between a cell and its neighbour. The material is redistributed, taking account of the elevation of the orthogonal surrounding cells (Fig. 6). The sediment metrics are then updated accordingly, including the total volume of material and the cell's elevation above a reference point. The rules defining the sediment redistribution are important parameters that can significantly alter the model outcomes and have therefore been thoroughly tested. The two most critical components are (1) the threshold angle between cells that instigates the redistribution method and (2) the frequency that the domain is analysed for these thresholds. Whilst the longshore sediment transport method is carried out along the shoreline on every timestep, activating the sediment distribution method at this frequency causes instability in the model and it is therefore activated less frequently. The threshold and frequency parameter values should be calibrated to allow sediment to be distributed without inducing sediment pilling or deep depressions forming in the domain. Similar techniques are widely implemented in landscape evolution models, such as SIBERIA (Willgoose et al., 1991) and GOLEM (Tucker and Slingerland, 1994) (Coulthard, 2001)."*

We agree that validation of the cross-shore distribution of sediment across the nearshore profile would be advantageous. However, for this we would need field data and field examples of which we do not have.

| 18 | I also miss a lot of discussion and review of recent existing models accounting for alongshore and cross-shore responses and accounting for changes in sea level, for example, Larson et al. (2016), Vitousek et al. (2017), Robinet et al. (2018), and Antolinez et al. (2019). |
|---|---|
| | **Thank you for your suggestions and for the list of references at the end of your response. We have edited the text in Section 1 starting at L51 that details some of the existing models and added these authors to the review. We recognize that this list is not exhaustive but contains some of the key models and texts relevant to the paper.**

 **L51:** *"In the field of coastal modelling, there is a gap for a two-dimensional coastal model that can simulate features such as spits, bars and beach migration along with a dynamic nearshore bathymetry and a variable water level but is parsimonious enough to enable short run times allowing us to answer research questions about coastal evolution at meso-spatiotemporal scales. Existing models with such scope, such as CEM, COVE and GENESIS (Hanson and Kraus, 1989; Ashton et al., 2001; Hurst et al., 2014), are limited to transporting sediment in one-dimension and represent the coastline simply as a line with little accommodation for the nearshore shape or bathymetry. This means the models are parsimonious and fast, but are limited in their application, for example, to investigate the effects of sea level rise on costal geomorphology. Hybrid shoreline change models such as COCOONED (Antolínez et al., 2019) and CoSMoS-COAST (Vitousek et al., 2017) calculate sediment transport in cross-shore and long-shore directions and can varying the water level in model but are transect based and do not include a dynamically evolving bathymetry. The LX-Shore model (Robinet et al., 2018) is cellular-based with longshore and cross-shore sediment transport calculations but has an equilibrium beach profile as in models such as CEM and COVE. In contrast to these longer-term models, finer scale models such as Delft3D (Lesser et al., 2004) can simulate coastal hydrodynamics and sediment transport processes in two- or three-dimensions, but their complexity and long model run times means investigating sea level rise responses over meso-timescales is presently impracticable."* |

| 19 | In the abstract the authors explain the model is suitable for evolving morphological features in time scales from 10 to 100 years, but any analysis is performed in these timescales. |
|---|---|
| | **The models were run over a simulated period of 3,000 years, to allow time for the model to spin-up, to reduce the potential influence of initial conditions, to allow sufficient time for the coastal systems to evolve to a state of relative stability and to generate a sufficient amount of data to analyse the behavior of the systems (see L172). Whilst this is the case, the results are applicable to timescales of 10 to 100 years and we agree that this should be better expressed in the paper. The following text has been edited at L176:**
 *"The models were run over a simulated period of 3,000 years, to allow time for the model to spin-up, to reduce the potential influence of initial conditions, to allow sufficient time for the coastal systems to evolve and to generate sufficient data from which quantitative and qualitative analysis could be completed. Whilst this is the case, the results are applicable to timescales of 10 to 100 years."* |

| 20 | I support the idea of changing the bathymetry, but what is the added value if wave transformations are still assuming parallel contours to the shoreline as in CEM? other models such as Robinet et al., 2018 already account for a scheme propagating waves in complex bathymetry and studies such as the one presented in Limber et al. (2017) proofs its importance. |
|---|---|

**This is a great point. With CEM2D we have made the topography 2D but retained a 1D approach to waves. This really comes down to the scope and parsimony of the model. Adding in wave transformations according to the changing bathymetry/topography would be a great addition and something we have thought about. However, the complexity and associated numerical overhead associated with this then restricts the spatial and temporal scope over which the model can be applied. Simply, we have chosen or line of simplification to be with a simple wave approach. CEM2D is not, nor intended to be a complete solution but by including a dynamically evolving topography and bathymetry, the model can be used to analyse how the profile of the coastal system evolves over time, the morphology of shoreline features and the existence of remnant features that are stored in the bathymetry. It was also necessary in order to account for a variable water level.**

**The advantage of having a dynamically evolving bathymetry also means that we can continue to increase the complexity of the model through the additional of processes that use metrics of the bathymetry, such as wave transformations.**

| 21 | I can read several times through the text the authors acknowledge certain model limitations and they propose to incorporate improvements in coming versions, why do not incorporate them now? (for example, lines 238-240) |
|---|---|
| | **We acknowledge that there are several limitations of the model highlighted in the manuscript. These have been included, as it is important to understand how the model works and what the limitations are. Model development is an ongoing process and we discuss these limitations, as areas we feel are important to focus on for future developments, that will require a significant amount of research to address.** |

| 22 | Certain Figures are not properly presented, for example Figure 10 and Figure 11 cut the model domain and shoreline shapes are not complete, Figure 9 has different color markers in the legend than in the subplots. |
|---|---|
| | **Figure 9 has been updated with the correct markers.** |
| | **Figure 10 and 11 have been reduced along the x-axis so that the entire matrix of results can be viewed on a single page. They have been reduced by cropping the periodic boundaries from the model domain, which are repeats of the central portion of the domain (shown). Removing these Sections, therefore, do not impeded the results.** |

---

## Referee Report (RR1)

**GMD-2019-197 REVIEW v2**

I am very sorry to communicate that I strongly recommend not considering this work for publication. I think you have got great ideas on how CEM could be improved, and you have implemented some of them, but this research is incomplete and presents several inconsistencies in the approach that at least need investigation. Presenting a new model requires validation. There are fundamental processes that I have the impression CEM2D could be messing up. The development of complex bathymetries but the non-inclusion of 2D wave transformation is inconsistent. The cross-shore profile shape evolution requires verification, modelling its evolution is not trivial. The use of a 2D topography scheme with an integrated alongshore sediment transport formula taking sediment from the cell adjacent to the shore is inconsistent, unless you adapt this shore-cell to cover the active zone (that would change in time), or unless you assume a constant profile shape (as in CEM). I encourage the authors to take my comments positively to reinforce their research, and that soon we can read a new resubmitted version of CEM2D with a thoroughly verification of the new processes that is accounting for.

Below my responses to author comments with more fundamental questions regarding their results and presentation of the paper.

11- CEM assumes a constant active cross-shore profile shape and the sediment transport is computed integrated in the surf-zone, for example with the CERC formula. Therefore, the sediment is added/removed horizontally. CEM2D claims to take the sediment from the numerical-grid cells in vertical, this is a great addition but brings inconsistencies when sediment transport is still computed with an integrated formula as in CERC and this sediment is only taken from the adjacent cell to the shoreline, this cell could not be representative of the active profile, could cut it in half, or be double. Therefore, it could introduce changes in the profile shape that would be artifacts of the model grid or the sediment redistribution scheme.

"The evolution of landforms is more gradual in CEM2D..." could you proof this is an improvement over CEM to match natural landform evolution?

12- Figure 17 shows that under sea level rise the landscape feature under the same wave climate is different, could you verify this is a natural behavior? For example, you could find a spot in the world where large scales features developed during a transgressive period, proof that with your model you can mimic the spatial scale and shape of these features, and with CEM the resultant feature would be different.

13- Thanks for the answer, I am curious to know which model is doing alright, could you proof your model is improving the results given by CEM? I do agree that CEM2D evolves different landscapes due to the redistribution scheme, but this could be a numerical artifact, it needs any kind of verification.

14- I am surprised, it is not a numerical instability, however it is a directional bias inherited from the model routines... in any case is a model artifact, I cannot read this in line 239. If CEM2D develops more complex topography and it is different than CEM results, please, proof that your model reproduces natural landscape evolution better than CEM. You could add in Section 5.2 the examples you mention and highlight the differences between CEM2D and CEM in comparison to the natural feature. Nowadays, you can easily obtain accurate wave climates around the world from ERA5 reanalysis (40 years dataset) so that you can compute A,U.

15- In science it is not enough with saying it, we need validation/proof that CEM2D mimics the natural landscape evolution. Please, validate that your model could be used for landscape prediction, validate that the shape, spatial scale, and temporal scale matches the natural behavior.

In section 5.2.1. CEM2D under A=0.55 and U=0.6 doesn't show any feature like the Carolinas, in any combination of A and U is not possible to develop large scale Cuspate Forelands with wave lengths longer than 50km or amplitudes greater than 10km.

15.r The cross-shore profile shape is of great relevance and modelling its evolution is not trivial, proof that CEM2D mimics the profile shape evolution.

16. Presenting a new model requires validation.

17. In case the model takes all the sand from the shore cell, if the shore cell is half the active profile width is taking too much sand from the vertical. I think it is inconsistent this numerical scheme with an integrated sediment transport formula. You need to take sediment at cell level…otherwise you introduce spurious bed level updates (that later you try to smooth out with your redistribution technique).

*"activating the sediment distribution method at this frequency causes instability in the model and it is therefore activated less frequently"*

Why do you think this happens?

19. It is not enough to say that the results are applicable to timescales of 10 to 100years, show what are the model outcomes at those timescales, do the results mimic shoreline rates found in nature? In general, you must validate that the model mimics the time scale of large-scale feature formation. What is the spin-up of the model?

20. Again, this is an inconsistency, you claim to develop complex bathymetries that is proven to affect shoreline feature development (Limber et al., 2017), but you don't account for changes in wave transformation. I think from your new Figure 17, this affects the results output by CEM2D.

---

## Author Response (AR2)

Author's Response to Reviewer Comments

Authors comments in **bold**, text added to the manuscript are in red.

We would like to thank all three reviewers for their constructive comments on this manuscript. We would like to address each of your comments and the changes we have made to the paper in response to your feedback. Within our responses, text highlighted in red is that which has been changed or added to the manuscript.

**Editor:**

Thanks for your patience. Could you please address the commends of Reviewers #2 and #3 in revised version of the manuscript? In particular I suggest that you provide more details in the model description and numerical schemes being used and include more model validation (as required by the GMD guidelines https://www.geoscientific-model-development.net/about/manuscript_types.html#item1)

Thanks. Stay safe and well

Lutz Gross, GMD Topical Editor

**Thank you for the overview and positive comments on our manuscript. We address all the comments to each of the reviewers below. With regard to the model validation issue you raise, we would like to bring to your attention that our paper is framed as a Development and Technical Paper, that according to the GMD guidelines should "provide an evaluation against standard benchmarks, observations, and/*or other model output* as appropriate". As such our new 2D model is evaluated against the performance of an existing 1D version of the model; this evaluation is similar but of course distinct from validation. We fully address this point in response 6 to Reviewer 2.**

**Reviewer 1:**

I am glad to see that the authors have made a considered and substantive effort to address my concerns and those of the other reviewer.

The power of CEM is in its demonstration of shoreline changes wrought by gradients in alongshore sediment transport. I think the major step still to come will be in marrying these changes in CEM2D v1.1 to the cross-shore profile with those gradients in alongshore flux, and seeing how that fully integrated consideration of the nearshore sediment-transport system affects modelled scenarios. (The authors allude to this caveat in the paragraph at L395.)

This work seems to me in the spirit of "geoscientific model development" – and relieving the assumption here of shoreline-parallel bathymetric contours is an important developmental step toward some potentially very interesting scientific questions.

**Reviewer 2:**

| 1 | 11 - CEM assumes a constant active cross-shore profile shape and the sediment transport is computed integrated in the surf-zone, for example with the CERC formula. Therefore, the sediment is added/removed horizontally. CEM2D claims to take the sediment from the numerical-grid cells in vertical, this is a great addition but brings inconsistencies when sediment transport is still computed with an integrated formula as in CERC and this sediment is only taken from the adjacent cell to the shoreline, this cell could not be representative of the active profile, could cut it in half, or be double. Therefore, it could introduce changes in the profile shape that would be artifacts of the model grid or the sediment redistribution scheme.

(**similar question:** 15.r The cross-shore profile shape is of great relevance and modelling its evolution is not trivial, proof that CEM2D mimics the profile shape evolution.)

"The evolution of landforms is more gradual in CEM2D…" could you proof this is an improvement over CEM to match natural landform evolution? |
| --- | --- |
|  | **In CEM, the change in horizontal fill of each shoreline cell is calculated based on how much sediment is moved and the distribution of this sediment across the shoreface assuming shore-parallel contours. Our approach is not so dissimilar from this but rather than distributing the volume of transported sediment across the shoreface assuming shore-parallel contours, CEM2D redistributes sediment across the profile using a steepest descent method that is compatible with the 2D profile. Similar steepest descent techniques are widely implemented in landscape evolution models, such as SIBERIA (Willgoose et al., 1991) and GOLEM (Tucker and Slingerland, 1994).**

**The CERC formula is usually used for uniform cross-shore profiles and whilst we technically have a dynamic profile, a threshold is used in the sediment distribution method which limits the slope angle that can be achieved. In doing this, the slope retains an average profile as can be seen in Figures 13, 16 and in the contours shown in Figures 11 and 14. Whilst this limits the dynamism of the bathymetric profile, it remains consistent with the use of the CERC formula in this iteration of the model. The methods employed here enable CEM2D to be a steppingstone towards a more complex representation of coastal evolution at this timescale and addition of vertical sediment storage was also necessary in order to model changes in the water level.**

**An alternate method for looking at more complex wave transformations was employed by Limber et al., (2017), who coupled the CEM with SWAN. SWAN carried out the complex and computationally expensive wave transformations over a non-uniform bathymetry and CEM used the transformed wave conditions in the CERC equation but retained a cross-parallel profile. This coupling could prove a useful concept for future iterations of CEM2D, with the added advantage that the bathymetry evolves in CEM2D, not just SWAN. This will however increase the computational time of the model, it's complexity and it's application to longer-term coastal change. In this iteration of the model we have taken a step towards achieving more complex coastal change without significantly increasing the complexity of bathymetric evolution of wave transformations. We recognise the limitations of this, but also highlight the advantages.** |

However, we note the point the review has made here and we have made the numerical schemes used for the sediment distribution method clearer in the manuscript, which now reads:

L135: *"Sediment flux is calculated using the same equations as CEM (Eq. 1) and threshold-determined upwind or central finite-difference techniques (Ashton et al., 2001; Ashton and Murray, 2006a), but because CEM2D represents sediment transport in two-dimensions, an alternate method for distributing sediment across the surf zone is used. Rather than assuming shore-parallel contours, material is dispersed across the surf zone based on a steepest descent method (Fig. 6). The method ensures that sediment is distributed across the active profile and remains consistent with transport calculations using the integrated CERC formula, but that there is dynamism in this process that takes into account the elevation of a cell and its neighbours that is consistent with the 2D representation of the domain."*

The sediment transport method (including sediment distribution across the surf zone) and wave climate representation are two elements within the model that we feel should be increased in complexity simultaneously, if either are to be altered. This is a significant task that would allow a more site-specific application of this model, but this is not the intention of this version of the model. This study was designed to demonstrate that the technical developments of CEM2D enable it to remain parsimonious and simulate fundamental shoreline shapes, as per the original CEM, but with added functionalities including the ability to vary the water level. We feel we have been able to do this in this manuscript. We do however take note of your comments and appreciate your thoughts on this topic; we endeavor to continue developing CEM2D for a range of applications and where we intend for a more site-specific application, we will be looking further into addressing the complexity of these components.

Ashton and Murray (2006) state that CEM was not designed to represent or predict the time evolution of any one specific coast or the specific development style of natural shoreline features; this also applies to CEM2D. This is not the principal purpose of these modelling exercises, although we indicate that we are assessing meso-scale behaviors, but it is rather to explore the relationship between shoreline feature types and wave climate characteristics. In response to your comments and to ensure that the reader is clear on these points we have now emphasized this point in the text:

L327: *"Whilst CEM and CEM2D are not designed to represent the temporal evolution of specific coastal environments and this metric should not therefore be compared between the models (Ashton and Murray, 2006a), we note that the evolution of landforms is more gradual in CEM2D."*

| 2 | 12 - Figure 17 shows that under sea level rise the landscape feature under the same wave climate is different, could you verify this is a natural behavior? For example, you could find a spot in the world where large scales features developed during a transgressive period, proof that with your model you can mimic the spatial scale and shape of these features, and with CEM the resultant feature would be different. |
|---|---|
| | We have added additional text to the manuscript to address this comment: |

**L399:** "Observing the effects of sea level rise on coastal features, including their ability to migrate with the shoreline or how their morphology changes at this temporal scale is challenging. Evidence of submerged shorelines and landforms that formed during transgressive periods can be removed by high energy waves and storm events, rapid migration of systems and by sediment transport that consumes or removes remnant features (Shaw et al., 2009). Notable submerged shorelines are found in the Bras d'Or Lakes, Nova Scotia, and are suggested to have been well-preserved by the rapid onset of sea level rise (Shaw et al., 2009). Tombolos, spits, cuspate forelands and barrier beaches are identifiable on multibeam sonar imagery in the Lakes down to -24 m, above the early Holocene water level at -25 m (Shaw, 2006). Evidence of enclosed bodies of water within cuspate forelands and the stranding of landforms at this lower sea level demonstrates in situ drowning and the preservation of landforms between -7 to -24 m evidences the ability of some landforms to migrate (Shaw, 2006; Shaw et al., 2009). Barrier islands and spits in the Bras d'Or Lakes are also found to rebuild at the proximal end of previously submerged landforms (e.g. Dhu Point and West Settlement) or migrate landward to form cuspate barriers in response to rising water levels (e.g. Goose Pond) (Taylor and Shaw, 2002)."

| 3 | 13 - Thanks for the answer, I am curious to know which model is doing alright, could you proof your model is improving the results given by CEM? I do agree that CEM2D evolves different landscapes due to the redistribution scheme, but this could be a numerical artifact, it needs any kind of verification. |
|---|---|
| | **We are not attempting to replicate any particular system in detail, so it is difficult to assess whether CEM or CEM2D is better at replicating any given specific natural systems (see also response number 1 and the changes made in the text). We have however qualitatively demonstrated that the fundamental shoreline shapes are represented better in CEM2D (see response number 14).** **What we have been able to show is that we can mimic fundamental shoreline shapes according to the driving wave conditions and evaluated these outputs against the CEM and natural systems. The intention of CEM2D is to continue using the model for this purpose, but it can obtain additional information and further exploratory capabilities with the addition of a variable bathymetry and ability to simulate a variable water level. We agree that model evaluation is an important step in model development, and we have addressed this point in response number 6, in line with GMD guidelines.** |

| 4 | 14- I am surprised, it is not a numerical instability, however it is a directional bias inherited from the model routines… in any case is a model artifact, I cannot read this in line 239. If CEM2D develops more complex topography and it is different than CEM results, please, proof that your model reproduces natural landscape evolution better than CEM. You could add in Section 5.2 the examples you mention and highlight the differences between CEM2D and CEM in comparison to the natural |
|---|---|

feature. Nowadays, you can easily obtain accurate wave climates around the world from ERA5 reanalysis (40 years dataset) so that you can compute A,U.

Our emphasis in this manuscript is not to compare the accuracy of shorelines shapes generated in CEM and CEM2D (see response number 3). We are showing that the model can simulate fundamental shoreline shapes according to the driving wave conditions, but with its added functionalities we can also use it to look at the dynamics of the surf zone and responses to sea level rise. We have evaluated the behaviours of the model against a number of case studies and standard benchmarks.

Having said that, in further response to your question, there are examples where CEM2D generates features that are more representative of natural systems compared to CEM. These examples include Benacre Ness, Long Point spit and Spurn Point. We have expanded on the text within the manuscript to reflect and highlight this point:

> L289: "Comparing these results to the planform morphology of sand waves found in natural systems, such as Benacre Ness in the UK which has PDF values of A=0.6 and U=0.8, demonstrates the ability of CEM2D to reflect the asymmetry of landforms formed under asymmetric wave climate conditions compared to CEM"

> L303: "Under all four potential wave climate conditions, reconnecting spit features form in CEM (Fig. 10), whereas in CEM2D (Fig. 11) either sand waves or reconnecting spits form depending on the combination of A and U values within the given ranges. Ashton and Murray (2007) suggest that the wave climate is favoured towards an asymmetry (A) of 0.8 along the entire spit and under these conditions, reconnecting spits form in CEM2D (Fig. 11), as per the natural system. The presentation of both sand waves and reconnecting spits in CEM2D would suggest that this model may be able to better represent the conditions found at Long Point Spit."

> L314: "Whilst CEM2D better represents the influence of low angle waves on coastal evolution at Spurn Point, it is of note that this is a complex feature which is influenced by conditions that could be having a greater impact on coastal evolution, including estuarine processes and dredging activities, that are not included in either CEM or CEM2D."

We have also noted where the CEM2D underperforms compared to CEM:

> L282: "Considering that all site-specific conditions controlling the evolution of capes are not represented in CEM2D or CEM, the models are able to predict a comparable shoreline type to that observed in this natural system although CEM2D overpredicts the directional skew."

Finally, the reference to model artifacts is now given in L259, onwards:
> "It is found that there is some directional bias in the source code that drives a longshore current independent of the wave climate conditions. This directional bias is more apparent in CEM2D and particularly where the wave climate is symmetrical (A = 0.5). It also drives some migration of the cuspate landforms downdrift, but a similar rate of movement is recorded in both CEM and CEM2D at 1.6 m and 1.7 m per year respectively. The directional bias is induced by calculations in the model that process from the left to the right of the domain. In future

| | |
|---|---|
| | **model versions, the routines will require updating which would also necessitate that sediment transport methods be altered accordingly.”** |

| | |
|---|---|
| **5** | 15- In science it is not enough with saying it, we need validation/proof that CEM2D mimics the natural landscape evolution. Please, validate that your model could be used for landscape prediction, validate that the shape, spatial scale, and temporal scale matches the natural behavior.

In section 5.2.1. CEM2D under A=0.55 and U=0.6 doesn't show any feature like the Carolinas, in any combination of A and U is not possible to develop large scale Cuspate Forelands with wave lengths longer than 50km or amplitudes greater than 10km. |
| | **The model has not been designed to predict landscape evolution in any particular location. This would require validation at the site-specific scale. Whilst this is something we envisage for future modelling development, this version is not designed for this purpose. It is rather designed to be exploratory and explore theories of high angle wave instability that we observe in nature.**

**We have noted the directional skew, but also shown in the model that symmetrical wave climates generate symmetrical features and asymmetric generate skewed features. This would theoretically explain the slight asymmetry in the Carolina Capes but we also note that the model overpredicts this skew due to directional bias. This example was also used by Ashton and Murray (2006) who noted that the timescale of cape development within CEM does not align with their formation, but that the model is able to demonstrate the role of high angle wave instability in their formation. The wavelength and amplitude of landforms in the model are restricted to the size of the domain. If therefore the intention was to model a specific system, the domain would be setup so that the boundaries would be sufficiently far enough to not influence the development of the landform. This was not our intention in this case.**

**According to the GMD guidelines, a Development and Technical Paper should "provide an evaluation against standard benchmarks, observations, and/*or other model output* as appropriate". We have provided such evaluation in Section 5 of the manuscript and have endeavored to strengthen these evaluations in response to other comments by the reviewers. In particular, we have built upon comparisons between observed natural features and outputs of CEM2D (e.g. Spurn Point and Long Point Spit Section 5.2.3), demonstrating the strengths of the model compared to CEM. We have also added further evaluation about the variable water level component in the model against observations of shoreline change in the Bras d'Or Lakes, Nova Scotia (Section 5.5).**

**Our paper therefore aligns with the spirt of GMD and highlights the behavior of the system and the model. Specific validation is not a requirement, but we do evaluate the model against CEM and we now highlight this in the text as outlined below.** |

| | |
|---|---|
| **6** | 16. Presenting a new model requires validation |
| | |

Truly validating models at this scale in this field is a significant challenge given that data is limited, although we strongly agree suitable testing and evaluation of model outputs is important.

GMD's guidelines for Development and Technical Papers are to "provide an evaluation against standard benchmarks, observations, and/*or other model output* as appropriate". There are no standard benchmarks or observations available for model performance so here we have *evaluated* the model against previous model output – that from the original CEM.

Our manuscript highlights that CEM2D builds on many concepts developed in the original CEM retaining many of the fundamental ideas and implementation of equations. Therefore, an important process in the *evaluation* of CEM2D was to ensure that it maintained the ability to simulate shoreline shapes as in CEM, that we have discussed in this manuscript. Here it is important to note that the original CEM and CEM2D were not designed to simulate any particular shoreline in detail, but as we have shown here, to connect theories of high angle wave instability to the shoreline shape and wave conditions of natural environments; we have used a number of examples in the manuscript to do this.

We have made our methods of evaluating the performance of CEM2D clearer, by adding the following text:

L73: "Validation of exploratory models like CEM2D is limited and particularly in this case, where there is a lack of data showing the evolution of coastal systems under changing wave patterns and water levels over such long time periods. CEM2D's performance is therefore here evaluated against 'standard' CEM simulation results from varying wave climates and directions."

L172: "To evaluate how CEM2D simulates coastal change, CEM2D was compared to CEM model outputs as well as to the behaviour and morphology of natural coastal environments. This provides both a check that the new model is able to represent natural systems as the original, but also to indicate where the added features (namely 2D operation) might change the model outputs. As the aim of this paper is to describe and highlight the technical developments of CEM2D we evaluate our simulation results against the original CEM outputs (as described subsequently). Full validation would require time series of bathymetric field data for the duration and range of wave climates and wave directions simulated, and this is not presently available. However, similar to how (Ashton and Murray, 2006) visually compare their simulation findings to coastal features including the Carolina Capes, we too compare our outputs to a series of examples."

L412: "The purpose of this study was to provide an overview of the development and application of CEM2D and its ability to represent coastal systems compared to other existing coastal evolution models of its kind. The behaviour of the model has been evaluated against results from the existing Coastline Evolution Model (CEM), upon which CEM2D has been built. Results have also been compared to accepted theories of coastal morphodynamics and to the behaviour of a number of natural coastal environments. These evaluation techniques have demonstrated that CEM2D is able to simulate shoreline instabilities in accordance with theories of high-angle wave instability, to mimic the behaviour of natural environments under given wave climate conditions and to generally reproduce the results of the original one-line

**CEM, although some differences are observed as discussed throughout** (Ashton and Murray, 2006a).”

L446: “We have described the structure of the model, outlined the governing mathematical equations, presented outputs from the sensitivity testing and **evaluated** CEM2D's ability to simulate the behaviour and evolution of coastal systems, **by comparing against other model results, theories of coastal evolution and natural systems**.”

| 7 | 17. In case the model takes all the sand from the shore cell, if the shore cell is half the active profile width is taking too much sand from the vertical. I think it is inconsistent this numerical scheme with an integrated sediment transport formula. You need to take sediment at cell level…otherwise you introduce spurious bed level updates (that later you try to smooth out with your redistribution technique). *"activating the sediment distribution method at this frequency causes instability in the model and it is therefore activated less frequently"* Why do you think this happens? |
|---|---|
| | **This question is similar to question 11, which we have responded to in response number 1 of this document.** **In response number 1, we have addressed concerns over inconsistencies with using the CERC formula with a non-uniform bathymetry. We have made further improvements to the way in which the sediment distribution scheme is described in the manuscript and we hope that this now makes the method much more transparent.** **To address the second point, instabilities due to moving too much sediment too quickly is a common problem in numerical modelling and is not unique to CEM2D. The two principal parameters for this method include the frequency that the method is employed and the threshold for movement. Both of these parameters were tested and were included in the sensitivity test to determine the most suitable values (see Section 5.1).** |

| 8 | 19. It is not enough to say that the results are applicable to timescales of 10 to 100 years, show what are the model outcomes at those timescales, do the results mimic shoreline rates found in nature? In general, you must validate that the model mimics the time scale of large-scale feature formation. What is the spin-up of the model? |
|---|---|
| | **The spin-up period is approximately 10 years, and this has been considered in the analysis of the results. The following text has been added to the manuscript to state this:** L189: “The models were run over a simulated period of 3,000 years, to allow time for the model to spin-up **(~10 years)**, to reduce the potential influence of initial conditions and to allow sufficient time for the coastal systems to evolve.” |

The exploratory nature of CEM2D is such that it is designed to look at behaviours the occur within millennial timescales. This is in accordance with the origins of the model and how CEM was designed and applied (Ashton, Murray and Arnault, 2001). For instance, Ashton and Murray (2006) noted that the timescale of cape development within CEM did not align with the formation of the Carolina Capes but nevertheless what was intended was to demonstrate that the driving wave conditions could lead to cape development (see also response number 5). We agree that this has not been expressed particularly well in the manuscript and have altered the text in-line with the timescales quoted by Ashton et al., (2001) that are clearer to interpret:

L29: "Simulating changes in coastal geomorphology up to millennial timescales and up to hundreds of kilometres"

| 9 | 20. Again, this is an inconsistency, you claim to develop complex bathymetries that is proven to affect shoreline feature development (Limber et al., 2017), but you don't account for changes in wave transformation. I think from your new Figure 17, this affects the results output by CEM2D. |
|---|---|
| | The study by Limber at al., 2017 computes wave transformations in SWAN, with only updating of the shoreline in CEM and is therefore note entirely comparable to other methods. |

The complexity of the bathymetric evolution, including cross-shore sediment transport processes and the complexity of the wave transformation equations should be updated concurrently. Whilst we have updated the representation of the bathymetry, it remains simplistic in the cross-shore diffusion of sediment and its maintenance of an average slope angle. There is a balance between increasing the complexity and increasing the efficiency and usefulness of a model and we are currently at this point. If we are to update the wave transformations, we feel we would also need to update the dynamics of the bathymetric profile. Whilst this is intended in future iterations, it is not included in this version of the model as we are attempting the retain its parsimonious nature.

**Reviewer 3:**

| 10 | Please present more details on the governing equations, variables, shoreline identification, transport schemes, diffusion schemes, etc. I think my biggest confusion of the model is role of the sediment fraction variable 'Fi'. In CEM, Fi=1 means a sandy cell, Fi=0 means an ocean cell, and 0<Fi<1 is a shoreline cell. This variable is very important because it allows the shoreline to have a "partial cell representation". The 0<Fi<1 value of Fi is used determine the location of the shoreline. And as Ashton & Murray (2006) put it "the fractional value within a cell represents the cross-shore extension of the sub-aerial shore" This fractional shoreline cell approach can be seen in Fig 4 of the current manuscript. This approach seem somewhat in contrast to the "full cell" shown in Fig 2 of the current manuscript. |
|---|---|
| | Thus my confusion is: does CEM2D use the variable Fi and a partial cell representation of the shoreline or not? It is very hard to tell from the current manuscript. From what I have read, I'm |

guessing not, because the wet or dry cells can be simply identified as those that are below or above sea level, respectively. The problem with the "full cell" representation (that is the lack of a partial cell identification) is that the shoreline angle only exist with an angular resolution of 45 degree increments. See figure 2 for example, where the angle of the shoreline on the left half of the figure is either zero or 45 degrees. The shoreline angle on Figure 4 of the current manuscript instead has a much more subtle variation in the shoreline angle. And, of course with longshore transport, the angle of the shoreline relative to the waves is critical to determining the transport (see Eq. 1).

Therefore, in my opinion, the text / figures greatly need to improve to clarify some critical details about the model. For example, Figures 2 and 4 are so similar, yet also seem to present contrasting representations of the shoreline.

**Additional details have been added to the manuscript regarding the numerical schemes in the model, including the transport and diffusion equation (see response number 1).**

**The reviewer is right that the Fi variable is not currently used in this version of CEM2D. However, we look two cells either side of a shore cell to determine the shoreline angle so the angle may be 0, 22.5, 45, 67.5 or 90 degrees. We agree that this can limit the resolution of the shoreline angle but not to the extent of 0, 45, 90 as above. We acknowledge that this is a limitation of our approach, but the results demonstrate that because the wave angle used in the calculations uses both the local shoreline *orientation* ($\theta$) *and the approaching wave angle* ($\varphi_b$), which varies on every iteration, a range of coastal landforms that evolve under various high-angle wave scenarios can be modelled.**

**We do however appreciate your discussion on this point and note that this is not clearly explained in the text. To make this clearer we have updated the text to reflect our response:**

L121: "The variable Fi is not used in CEM2D to represent the partial horizontal fill of sediment, rather, each cell contains values for depth of sediment to the continental shelf, elevation of sediment above the water level or depth of water (Fig. 4b). Having these additional values of sediment fill in the vertical enables CEM2D to represent two-dimensional coastlines with greater topographic detail compared to the original CEM, as illustrated in Fig. 4."

L128: "The boundary between wet and dry is used to locate the shoreline (Fig. 5), using the same shoreline search technique as CEM. Once the shoreline is located, as per CEM, Linear Wave Theory is used to transform the offshore wave climate and the CERC formula to calculate sediment flux between the one-line shoreline cells (Equation 1). The limitations of not calculating the horizontal sediment fill of each cell (Fi) influences the sediment transport equations, by reducing the angular resolution of the local shoreline. However, as shown in the results, the model remains capable of simulating fundamental shoreline shapes."

**We understand the confusion with Figure 2 and Figure 4 and we have made alternations to the figure placements to ensure that it is clear that the original Figure 2 (now Figure 5) is purely a reference to CEM2D and the original Figure 4 (now Figure 3) is for CEM. The rest of the figure numbers have also been updated accordingly. The original Figure 4 (now Figure 3) caption has also been updated:**
"Plan-view schematic of CEM" as opposed to "Plan-view schematic of CEM2D"

| 11 | There is no mention of potential physical / numerical instability of the model, unlike in Ashton & Murray (2006). In CEM, the numerical scheme to shift between a central scheme and an upwind scheme (when a critical angle of 42 degrees is exceeded) is key to the model's stability. Are similar schemes used in CEM2D? This should be discussed. |
|---|---|
| | **The same scheme is employed in CEM2D and we have added additional details to the manuscript to detail this, with reference to Ashton and Murray (2006):**

L87: **"The angle of the deep-water wave crest and local shoreline orientation determines the direction of sediment transport between cells. If the local relative wave angle is less (greater) than the angle which maximises sediment transport, sediment flux is calculated using a central (upwind) finite-difference technique (Ashton et al., 2001; Ashton and Murray, 2006a)."**

L135: **"Sediment flux is calculated using the same equations as CEM (Eq. 1) and threshold-determined upwind or central finite-difference techniques (Ashton et al., 2001; Ashton and Murray, 2006a),** but because CEM2D represents sediment transport in two-dimensions, **an alternate method for distributing sediment across the surf zone is used. Rather than assuming shore-parallel contours, material is dispersed across the surf zone based on a steepest descent method (Fig. 6). The method ensures that sediment is distributed across the active profile and remains consistent with transport calculations using the integrated CERC formula, but that there is dynamism in this process that takes into account the elevation of a cell and its neighbours that is consistent with the 2D representation of the domain."** |

| 12 | Like the previous reviewers, I found myself wondering how the longshore transport redistributed in the cross-shore, and thus how the offshore bathymetry changes in response to growing shoreline perturbations? (In CEM this is handled by extrapolating the sub-aqueous shoreface slope away from the shoreline) I assume that this is handled only via the diffusion scheme, but it is not obvious that this is the case in the manuscript. |
|---|---|
| | **We appreciate that both reviewers have made reference to the sediment distribution method and so we have added an additional section to explain this method in more detail. Please see response number 1 for further details.** |

| 13 | I was quite surprised to see the strong difference between CEM and CEM2D in Figures 10, 11, and 18. Why are they so different if this update is still relatively straightforward? (As an aside, I recommend using the same color scheme to the CEM results (e.g., blue and tan instead of gray, even though there is no gradation due to topography in CEM) as for the CEM2D results. This would greatly improve the readers ability to compare the results. |
|---|---|
| | **The color scheme used in Figure 10 for CEM results has been updated to match results of CEM2D in Figure 11.** |

Although we have retained the core of CEM and the ability to simulate fundamental shoreline shapes, the additional functionalities in CEM2D are relatively complex in terms of the representation of the domain and sediment distribution methods. This leads to differences in the outputs, although the model is still able to replicate the relationship between high angle wave instability and coastal landform evolution. We have highlighted throughout the manuscript where differences are observed between CEM and CEM2D and have added further detail for clarity:

L71: "We describe in full the model's operation and parameterisation, and compare the model outputs to the original CEM, illustrating some similarities in model outputs but also key differences that are due to the improved two-dimensional representation of the coastline and sediment transport processes."

L269: "In CEM2D a greater distinction is made between reconnecting and flying spits due to the increased complexity of CEM2D's sediment handling and distribution methods. The distribution method allows sediment accumulations to be detached from the continuous shoreline without becoming static and so transport across the entire domain, including on the lee side of a spit, is less limited."

L282: "Considering that all site-specific conditions controlling the evolution of capes are not represented in CEM2D or CEM, the models are able to predict a comparable shoreline type to that observed in this natural system although CEM2D overpredicts the directional skew."

L289: "Comparing these results to the planform morphology of sand waves found in natural systems, such as Benacre Ness in the UK which has PDF values of A=0.6 and U=0.8, demonstrates the ability of CEM2D to reflect the asymmetry of landforms formed under asymmetric wave climate conditions compared to CEM."

L307: "The presentation of both sand waves and reconnecting spits in CEM2D would suggest that this model may be able to better represent the conditions found at Long Point Spit."

L313: "However, in CEM2D these features fluctuate between spits and sand waves owing to the strong longshore current generated by the low angle waves and high asymmetry. Whilst CEM2D better represents the influence of low angle waves on coastal evolution at Spurn Point, it is of note that this is a complex feature which is influenced by conditions that could be having a greater impact on coastal evolution, including estuarine processes and dredging activities, that are not included in either CEM or CEM2D."

L319: "The spatial scale of shoreline features differs between results from CEM and CEM2D. Metrics from the end of each run, as shown in Fig. 10 and Fig. 11, show larger features evolve in CEM2D in six of the simulations. The larger features evolve under wave climate conditions where A = 0.6, U = 0.55-0.65 (sand waves), where A = 0.7-0.8, U = 0.7 (flying spit) and where A = 0.9, U = 0.75 (flying spit) and smaller features evolve in the remaining nineteen simulations. However, each run terminates at a different timestep and a comparison of results at the earliest termination for each pair of simulations shows that in all but one of the runs (A = 0.6, U = 0.55), the features are smaller and less developed in CEM2D than the CEM (Fig. 18)."

L327: "**Whilst CEM and CEM2D are not designed to represent the temporal evolution of specific coastal environments and this metric should not therefore be compared between the models (Ashton and Murray, 2006a), we note that** the evolution of landforms is more gradual in CEM2D. This is likely as a result of differences in the representation of the domain and in the distribution of sediment."

L335: "Highlighted above are differences in results between CEM and CEM2D and in particular, the complexity of results generated in CEM2D due to the addition of a dynamically evolving profile. In nature, the features discussed evolve at different rates and to different spatial scales depending and in order to use CEM2D to investigate such systems, parameters in the model including the threshold and frequency of sediment distribution should be adjusted to suit the specific environment studied and the rates at which these features form."

L413: "**The behaviour of the model has been evaluated against results from the existing Coastline Evolution Model (CEM), upon which CEM2D has been built. Results have also been compared to accepted theories of coastal morphodynamics and to the behaviour of a number of natural coastal environments. These evaluation techniques have demonstrated that CEM2D is able to simulate shoreline instabilities in accordance with theories of high-angle wave instability, to mimic the behaviour of natural environments under given wave climate conditions and to generally reproduce the results of the original one-line CEM, although some differences are observed as discussed throughout** (Ashton and Murray, 2006a)."

| 14 | I know that CEM and CEM2D are typically applied in high wave angle, high wave asymmetry conditions (U,A>0.5), but do these models work when in low wave angle, low wave asymmetry conditions (U,A<0.5)? These low wave angle, low wave asymmetry conditions (U,A<0.5) persist for most coastlines, in general. I would like to see the behavior of this model in a more ordinary test case such as the diffusion of a Gaussian shaped shoreline bump to evaluate the differences between CEM and CEM2D, rather than going straight to the cases of flying spits. |
|---|---|
| | **The model has been designed and its application intended for high wave angle environments. It is already being applied for symmetrical (A = 0.5) and asymmetric (A > 0.5) wave climates. It would be interesting to apply the model to low wave angle environments in the future, but it wasn't the intention of his study. Thank you for the suggestion, we have taken this on board for future work.** |

[revised manuscript text omitted]

---

## Author Response (AR3)

**Author's Response to Reviewer Comments**

Authors comments in **bold**, text added to the manuscript are in red and line numbers refer to those in the track changed document.

We would like to thank reviewer 2 for their constructive comments on this manuscript. We would like to address each of your comments and the changes we have made to the paper in response to your feedback. Within our responses, text highlighted in red is that which has been changed or added to the manuscript.

**Editor:**

Thanks for your patience.

Reviewer #2 has asked for some more clarifications. I would like to highlight his comments on the different results produced by CEM and CEM2D. Under this light, it could be worthwhile to discuss when and where CEM2D would be the preferred option and when it would be appropriate to stay with CEM.

Thanks.

Lutz Gross

GMD Topical Editor

**Thank you for your response, we have addressed comments from Reviewer 2 below. We have added some additional details related to the numerical schemes used in the model, particularly relating to the calculation of the local shoreline orientation and avalanching scheme. We have also emphasized where CEM2D or CEM would be the preferred model, building on analysis throughout the manuscript where we have discussed and explained differences in model outputs.**

**Reviewer 2:**

| 1 | In my previous round of comments, I recommended: |
|---|---|
| | "Please present more details on the governing equations, variables, shoreline identification, transport schemes, diffusion schemes, etc." |
| | |
| | Unfortunately, I feel that I must ask for that again. In particular, I am still a bit unclear on the calculation of the shoreline angle, and I would like to see some more of the numeric/algorithmic details. For example, in response to my previous comment (i.e., comment #10), the authors mentioned: "the shoreline angle so the angle may be 0, 22.5, 45, 67.5 or 90 degrees". |
| | |
| | This is certainly better than 0, 45, 90 degree increments (like I thought was the case previously), but I think the method could still be improved to be better than 0, 22.5, 45, 67.5 or 90 degrees. If you look three grid points on either side, then would this give you 15 degree increments? And, if you look four grid points on either side, then would this give you 10 degree increments? I'm guessing that 4 grid points on either side (i.e., 10 degree increments) would probably be sufficient. If you |

have any mathematical details to present on this, then I would greatly appreciate seeing more of the detail.

**Whilst we agree that by increasing the number of grid points either side of the cell in question would increase the range of degree increments able to be calculated, though this raises a series of issues itself. Changing the number of cells also effectively smooths the angle and this may have positive and negative impacts in that smaller variations within coastal cell positions may be ignored (smoothed) or have a disproportionate impact. Given then the 'on-off' 'wet-dry' nature of cells being used to count the shoreline, it may well be that smoothing or accounting for the shoreline angle over a greater length of cells may be advantageous but of course, too long a length may result in features being effectively ignored thereby a constant angle coming into effect and the impact of shoreline angle being made redundant. The smoothing or not effect, will also depend upon the spatial resolution being used in the model. This, in effect, becomes another parameter that would be required to be taken into account in any sensitivity testing. At present, we feel the current method, as used in CEM, generates enough variation in shoreline angle to capture the coastal dynamics we aim to simulate. It is, of course, something that could be investigated in future developments of the model.**

**We appreciate your comments, however, and have added some additional text to make the calculation of the local shoreline orientation clearer and be upfront if this represents any limitation. We feel that this additional text, along with the existing explanation and figure (5), the manuscript sufficiently describes how the model works and we hope that this is satisfactory:**

> **L131: "The local shoreline orientation is identified by computing the angle between a shoreline cell and two neighbouring shoreline cells. This forces the shoreline angle to be either 0, 22.5, 45, 67.5 or 90 degrees."**

**To try and keep the paper concise and to the point, we have not elaborated with the discussion points on smoothing above. However, this comment (and the above discussion) will be in the online published review and therefore available as a point of reference.**

**We have also added further explanation and equations to the manuscript, detailing the avalanching technique. Details of this are given in response number 3.**
* * *
| 2 | It appears the modeled shorelines on Figures 14 and 15 are fairly "staircased" or jagged. Can you please comment on whether this issue of limited angular resolution is contributing to that jaggedness. Overall, CEM seems to have a much smoother representation of the shoreline than CEM2D. |
|---|---|

**The reason for the staircase appearance that the reviewer refers to in Figures 14 and 15, is related to the representation of sediment as vertical fill in cells as opposed to horizontal (see Section 3). In CEM the partial horizontal fill of sediment in a cell is recorded in the outputs giving it a smoother representation of the shoreline. As discussion in response number 1, computing the horizontal fill of each shoreline cell is part of the ongoing development of CEM2D and it is intended to appear in future iterations of the model. Having partial horizontal fill will give CEM2D a smoother representation of the shoreline. We have added a line in the text to explain this:**

| | |
|---|---|
| | **L136: "and it can also lead to a more irregular representation of the shoreline."** |

| 3 | Previously, I also asked a question about "how the longshore transport redistributed in the cross-shore". The authors mention a "steepest descent method" (which means something else to me, i.e., an optimization method). I think more algorithmic details are still needed on this. It is clear qualitatively what they are doing in Figure 6, but it's not clear quantitatively what the method is doing, and thus I recommend more details, equations, etc. In the context of beach erosion models, I would maybe call this an "avalanching scheme" after Roelvink et al., (2009) "Modelling storm impacts on beaches, dunes and barrier islands" – Coastal Engineering. |
|---|---|
| | **Thank for referencing the work of Roelvink et al., (2009). The avalanching method used in XBeach are somewhat similar to the methods we use in CEM2D and so we have taken up your suggestion of using the term "avalanching" instead. We agree that this is perhaps a better description of the method used in CEM2D and have therefore updated the text accordingly:**

**L142: "Rather than assuming shore-parallel contours, material is dispersed across the surf zone based on an avalanching scheme that is somewhat similar to that used in other coastal evolution models (e.g. XBeach (Roelvink _et al._, 2009))(Fig. 6)."**

**We have also added some additional details to the description, as requested:**

**L151: "To carry out this redistribution procedure, an algorithm sweeps the entire model domain and identifies where a critical angle has been exceeded between a cell and its neighbour (Eq. 3).**
$$\frac{\Delta_z}{\Delta_w} > m_{cr} \qquad\qquad\qquad\qquad\qquad\qquad\qquad\qquad (3)$$
**where z is depth, w is cell width and $m_{cr}$ is the critical slope. The material is then redistributed amongst the orthogonal surrounding cells until the critical slope angle is no longer exceeded (Fig. 6)."** |

| 4 | Validation: I agree with the other reviewers comment that "Presenting a new model requires validation" and that "we need validation/proof that CEM2D mimics the natural landscape evolution." I also agree with authors' approach to try to validate against existing models, in the absence of a better validation test case. Shorelines with high-angle wave instabilities are quite a rare natural phenomenon, and it's even less likely that good data exists on them. I wish I could offer any suggestion to resolve this, but I really don't have a good solution. I'm guessing that I would leave it up to the editors to mediate.

The biggest problem that I'm still struggling with, in terms of the model validation, is why the results in CEM and CEM2D in Figures 10 and 11 look so drastically different, to me. Yes, they both capture some degree of the high wave angle instability features, but I am not entirely satisfied with the authors response in comment #13. To me, if the validation of CEM2D hinges on is ability to reproduce results similar to CEM, then there is perhaps still some work to be done on this front, because they still seem to be producing very different behaviors. |
|---|---|

We really appreciate the reviewer's acknowledgement of the limitations associated with validating a model of this kind and of our appropriate approach to this difficult task. Further development of CEM2D will continue to evaluate model performance not only against the behavior of other models like CEM but against natural systems as more data becomes available.

We have explored in the manuscript where model outputs differ between CEM and CEM2D. The results suggest that CEM2D better represents natural features in increasingly asymmetric wave climates compared to CEM, measured against example case study sites throughout the manuscript. For instance, CEM appears to perform better under a symmetrical wave climate (e.g. capes), and CEM2D performs better when simulating highly asymmetric wave climates (e.g. spits). As noted in the paper, the avalanching scheme in CEM2D allows sediment accumulations to be detached from the continuous shoreline without becoming static (L.280). CEM2D is therefore able to better represent these highly complex spit features that evolve with sediment accumulations that do detach from the continuous shoreline. We have included some additional text in the manuscript to discuss where CEM or CEM2D would be the preferred model option, based on this analysis:

L291: "Considering that all site-specific conditions controlling the evolution of capes are not represented in CEM2D or CEM, the models are able to predict a comparable shoreline type to that observed in this natural system. However, CEM2D overpredicts the directional skew and so CEM may be the preferred option in this instance."

L300: "demonstrates the ability of CEM2D to reflect the asymmetry of landforms formed under asymmetric wave climate conditions compared to CEM; CEM2D may, therefore, be the preferred model in this instance."

L315: "Ashton and Murray (2007) suggest that the wave climate is favoured towards an asymmetry (A) of 0.8 along the entire spit and under these conditions, reconnecting spits form in CEM2D (Fig. 11), suggesting that CEM2D may be there preferred tool to use in this conditions."

L427: "These evaluation techniques have demonstrated that CEM2D is able to simulate shoreline instabilities in accordance with theories of high-angle wave instability, to mimic the behaviour of natural environments under given wave climate conditions and to generally reproduce the results of the original one-line CEM, although some differences are observed as discussed throughout (Ashton and Murray, 2006a). In particular, the results show that CEM2D shows increasing model performance with increasing wave asymmetry compared to CEM. This is likely due to its ability to handle detached sediment accumulations that form during the evolution of reconnecting and flying spits, under these wave conditions. It may, therefore, be more appropriate to use CEM2D over CEM when modelling environments with asymmetric wave climates, but CEM where wave approach is highly symmetrical."

Changing the dimensionality of the model and enabling dynamic cross-shore evolution in CEM2D makes it the preferred model to use where the evolution of the bathymetry, including (for instance) morphological inheritance, is important or of interest:

L438: "Importantly, restructuring and increasing the dimensionality of sediment transport in the model allows us to explore how the profile of the coastal system changes with the shape of the shoreline, as well as concepts such as morphological inheritance. Where this is considered particularly important or of interest, CEM2D would be the preferred model to use over CEM."

The ability to model a variable water level in CEM2D offers an advantage over CEM, particularly when considering the impacts of sea level rise over the timescales these models are intended for:

L457: "A key component of CEM2D is its variable water level, which offers an added advantage over the use of CEM particularly when considering the impacts of sea level rise over the timescales these models are intended for."